

# How do convective cold pools influence the boundary-layer atmosphere near two wind turbines in northern Germany?

Jeffrey D. Thayer[1], Gerard Kilroy[1], and Norman Wildmann[1]

[1]German Aerospace Center (DLR), Institute for Atmospheric Physics, Oberpfaffenhofen, Germany
**Correspondence:** Jeffrey D. Thayer (jeffrey.thayer@dlr.de)

**Abstract.** With increasing wind energy in the German energy grid, it is crucial to better understand how particular atmospheric phenomena can impact wind turbines and the surrounding boundary-layer atmosphere. Deep convection is one source of uncertainty for wind energy prediction, with the near-surface convective outflow (i.e., cold pool) causing rapid kinematic and thermodynamic changes that are not adequately captured by operational weather models. Using 1-minute meteorological mast and remote-sensing vertical profile observations from the WiValdi research wind park in northern Germany, we detect and characterize 120 convective cold pool passages over a period of 4 years in terms of their temporal evolution and vertical structure. We particularly focus on variations in wind-energy-relevant variables (wind speed and direction, turbulence strength, shear, veer and static stability) within the turbine rotor layer (34-150 m height) to isolate cold pool impacts that are critical for wind turbine operations. Near hub-height (92 m) during the gust front passage, there are relatively increased wind speeds up to +4 m s$^{-1}$ in addition to the background flow, a relative wind direction shift up to +15°, and increased turbulence strength for a median cold pool. Given hub-height wind speeds lying within the partial load region of the power curve for the detected cases, there is an increase in estimated power of up to 50% which lasts for 30 minutes. We find a 'nose shape' in relative wind speeds and $\theta_v$ at hub-height during gust front passages, with larger wind direction changes closer to the surface. This manifests as asymmetric fluctuations in positive shear, negative veer, and stability across the rotor layer, with relative variations below hub-height at least twice as large compared with above hub-height and temporarily opposite signs for stability that have complex implications for turbine wakes. Doppler wind lidar profiles indicate that kinematic changes associated with the gust front extend to a height of 650-700 m, providing an estimate for cold pool depth and highlighting that cold pool impacts would typically extend beyond the height of current wind turbines. After the cold pool gust front passage, there is gradually increasing stability, with a decrease in $\theta_v$ to -2 K, a gradual decrease in turbulence strength, and faster recovery of wind speed than wind direction.

## 1 Introduction

With the share of wind energy in the German power grid increasing from 1.7% in 2000 to 34% over the first half of 2024 (BMWK, 2024; Fraunhofer ISE, 2024), the density of wind turbines across Germany is increasing. This indicates that smaller-scale phenomena can impact a greater number of wind turbines, and that the German energy sector is becoming more susceptible to shorter-duration power fluctuations. The stability and balancing of the German electrical grid operates on lead





times down to 5 minutes, which increases the utility of minute-scale wind power forecasting for energy trading and wind turbine control (Wurth *et al.*, 2019). Despite the identified importance of short-term wind power forecasting, there are continuing uncertainties with power forecasts. The median annual wind energy production still tends to be overpredicted globally, with uncertainties of ∼6% (Lee and Fields, 2021). Over Germany, larger day-ahead power forecast errors have been associated
with various meteorological events, with a 1-3% installed capacity day-ahead forecast error in any given month indicating a wide-range of meteorological causes possibly affecting the forecasting errors (Steiner *et al.*, 2017). Meteorological impacts on power production are therefore one noteworthy consideration for limiting the power prediction uncertainty, with reductions in wind speed ($U$) uncertainty having an out-sized improvement on predicted power ($P$) since $P \sim U^3$ (Lee and Fields, 2021). As such, improving understanding of short-term low-level wind speed variations associated with meteorological events is crucial
for reducing short-term wind power forecasting uncertainties.

Rapid changes in the wind profile over the lowest few hundred meters of the atmospheric boundary layer (ABL) on timescales ranging from a few minutes to a few hours (e.g., wind ramps) are an important driver of variations in wind power. Wind ramps can be produced by a range of meteorological phenomena, including mid-latitude cyclones, low-level jets, sea breezes, and deep convection, and therefore wind ramps can occur across a range of timescales and spatial scales (Gallego-
Castillo *et al.*, 2015). Past work on wind ramps over Europe has largely focused on the turbine impacts without providing a detailed verification of the meteorological cause of the wind ramps (Vincent *et al.*, 2011; Kelly *et al.*, 2021; Lochmann *et al.*, 2023). When meteorological causes of wind ramps have been identified, the focus was on synoptic-scale weather phenomena (Valldecabres *et al.*, 2020) or specifically mid-latitude cyclones (Steiner *et al.*, 2017). Steiner *et al.*. (2017) additionally found that 18% of analyzed wind ramps could be related to convective systems or unresolved downward mixing, but these cases were
not investigated in detail and deeper evaluation would have been limited by the ∼7 km horizontal resolution of the numerical weather prediction (NWP) model used in their study. There remains limited work concerning wind ramps linked to convective events over Europe.

In this regard, quantification of short-term thunderstorm impacts using wind-energy-relevant variables, and specifically related to thunderstorm near-surface outflow, has rarely been performed. In brief, mature-stage deep convection produces a
precipitation-induced downdraft with descending cold air that spreads outward once reaching the surface (Byers and Braham, 1948). This 'cold pool' has a low-level outflow boundary at its periphery known as a gust front, which is typically characterized by increased horizontal wind speeds, a wind direction shift, and increased turbulence relative to the background environment (Benjamin, 1968; Goff, 1976; Wakimoto, 1982; Droegemeier and Wilhelmson, 1987). As thunderstorm gust front winds tend to maximize within the height range of the turbine rotor layer (Lombardo *et al.*, 2014; Gunter and Schroeder, 2015; Canepa
*et al.*, 2020) and thunderstorm ramp-up events typically occur on timescales from 1-5 minutes (Lombardo *et al.*, 2014), gust fronts are simultaneously crucial for assessing short-term wind turbine-related impacts and under-resolved by NWP models both spatially and temporally. This limitation is potentially exacerbated by the fact that NWP models tend to underestimate extreme winds due to mesoscale smoothing, with this effect especially pronounced on the sub-hourly timescale (Larsen *et al.*, 2012). High-frequency kinematic observations encompassing the turbine rotor layer are thus a promising tool for accurately
characterizing rapid variations in wind power caused by convective cold pools and their associated gust fronts.





The gust front is not the only aspect of convective cold pools relevant to wind energy applications. Near-surface thermal stability changes in the ABL, such as those that occur after a cold pool passage, also have notable implications for variations in wind turbine power generation. A convective (stable) boundary layer is associated with higher (lower) turbulence (Stull, 1988), which contributes to the amount of turbulent mixing in the wind turbine wake region. Decreased wind speed deficits (i.e., faster wake recovery and shorter wake length) and greater turbulence downstream of wind turbines occur in convective conditions compared with a neutral boundary layer (Zhang *et al.*, 2013; Iungo and Porte-Agel, 2014; Abkar and Porte-Agel, 2014), generally leading to decreased power deficits for wind farms depending on turbine spacing (Hansen *et al.*, 2012). In short, the ABL stability affects the turbulence strength, which then influences how quickly the turbine wake recovers (Wildmann *et al.*, 2018) and whether the turbine wake impacts power generation of downstream turbines. With short-term increases in turbulence accompanying the cold pool gust front (e.g., Droegemeier and Wilhelmson, 1987; Lombardo *et al.*, 2014; Canepa *et al.*, 2020) and short-term increases in near-surface stability occurring within the cold pool interior, passages of convective cold pools over wind turbines provide a unique opportunity to examine the complex interactions between ABL stability regimes and wind power production.

With a particular focus on convective cold pool impacts within the turbine rotor layer, we investigate short-term kinematic and thermodynamic variations near two utility-scale wind turbines within the WiValdi research wind farm[1] in northern Germany from 2020-2024. The flat terrain and vast array of in-situ and remote sensing instrumentation available at this research site provide a unique opportunity for isolating and characterizing convective cold pool impacts relevant for wind energy. We extend the WiValdi climatological work of Wildmann *et al.*(2022) by highlighting thunderstorm-related environmental effects, with this region of northern Germany notably having a relatively high frequency of convective cells (Wilhelm *et al.*, 2023) and density of wind turbines (Bouchard and Romanic, 2023). Given the 1-5 minute timescale of thunderstorm ramp-up events and the importance of minute-scale forecasting for the German wind power sector, we utilize high-frequency (1-minute) measurements from meteorological mast, Doppler wind lidar velocity azimuth display (VAD) scans (Wildmann *et al.*, 2020), and microwave radiometer instrumentation at WiValdi to analyze the near-surface environmental changes associated with cold pools. Radar composite datasets from the German Meteorological Service (DWD) provide verification of the parent convective cells. Using summertime cold pool studies focused on the Netherlands (Kruse *et al.*, 2022) and Hamburg (Kirsch *et al.*, 2021; 2024) as a foundation, we examine year-round cold pool events with particular emphasis on wind-energy-relevant variables. The primary research questions guiding this work are as follows:

- How much does the wind speed, wind direction, and temperature change in the ABL during observed convective cold pool passages near the WiValdi wind turbines?

- To what extent do wind-energy-relevant variables (wind speed and direction, turbulence strength, wind shear, wind veer, static stability) change throughout the turbine rotor layer during these convective cold pool passages?

- Is there a significant variation in estimated wind power production at turbine hub-height during these convective cold pool passages?

---

[1]More information can be found on the official WiValdi wind park website (www.windenergy-researchfarm.com).



Section 2 outlines are observational datasets, instrumentation, and cold pool detection methodology. Sections 3 and 4
describe the temporal evolution and vertical structure of cold pool characteristics, respectively. Section 5 highlights changes
throughout the turbine rotor layer during cold pool passages. Section 6 contains our conclusions.

## 2 Data and Methodology

### 2.1 WiValdi instrumentation

The WiValdi research wind farm operated by the German Aerospace Center (DLR) in Krummendeich, Germany is located
in coastal flatlands near the mouth of the River Elbe by the North Sea (Fig. 1a). This large-scale research facility contains two
4.2-MW Enercon E115 wind turbines (Fig. 1b) that have hub heights of 92 m and rotor diameter $D$ of 116 m, are separated
by approximately 500 m (i.e., 4.3 rotor blade diameters = $4.3D$), and are aligned along the primary west-southwesterly wind
direction (Wildmann *et al.*, 2024). One 150-m meteorological mast located 2 $D$ in front of the westernmost turbine provides
inflow conditions at various heights throughout the turbine rotor layer, while 3 meteorological masts in between the turbines
provide detailed observations of the turbine wake. Using 1-minute mean measurements, the inflow meteorological mast obser-
vations, which are available starting from November 2022, will be used in this study to validate the remote sensing vertical
profiles and characterize the convective cold pool impacts within the rotor layer (i.e., 34-150 m). As a measure for turbulence,
1-minute turbulent kinetic energy (TKE) dissipation rate $\varepsilon$ is calculated from the second-order structure function according to
Bodini *et al.*. (2019) using ultrasonic anemometer observations that have a temporal frequency of 20 Hz. We use this variable
to characterize turbulence strength since it can be calculated for short time periods and can thus provide information about
rapid changes during gust front passages.

Vertical profiles of wind speed and direction are provided by a Leosphere WindCube 200S pulsed Doppler wind lidar that
was installed on 4 November 2020 approximately 400 m to the east of the easternmost turbine. This lidar has a pulse length of
200 ns, a measurement range for radial velocities between -30 to 30 m s$^{-1}$, and a minimum observing line-of-sight distance
of 100 m (two times the minimum range gate length). Radial velocity retrievals from a Doppler wind lidar are more greatly
attenuated by precipitation and low cloud cover (e.g., fog).

This lidar has the capability to perform horizontal plan-position indicator (PPI) scans, vertical range-height indicator
scans, and fixed direction scans. A sub-category of PPIs are conical scans at a constant elevation angle, known as Velocity-
Azimuth Display (VAD) scans, which allow for determinations of the horizontal wind components using the so-called VAD
technique. We perform VAD scans at two different elevation angles: 35.3º and 75º. The 75° VAD scan (VAD75) has a higher
vertical range and a smaller horizontal footprint, whereas the 35.3° scan (VAD35) can be used to determine TKE (Kropfli,
1986; Wildmann *et al.*, 2020). In both cases, an averaged profile of the 3D wind vector is measured horizontally across the
cone at each range gate (i.e., every height), with a complete scan lasting ~72 s. Each VAD scanning pattern is run for 30
minutes every hour, with 25 scans per elevation angle. Given a minimum measurement height of 57 m and 96 m for the VAD35
and VAD75 scans respectively, we obtain 'hub-height' lidar winds at 100 m to have consistent coverage during both scanning
patterns. Lidar data is linearly interpolated to 1-minute temporal resolution throughout the vertical profile to capture rapid



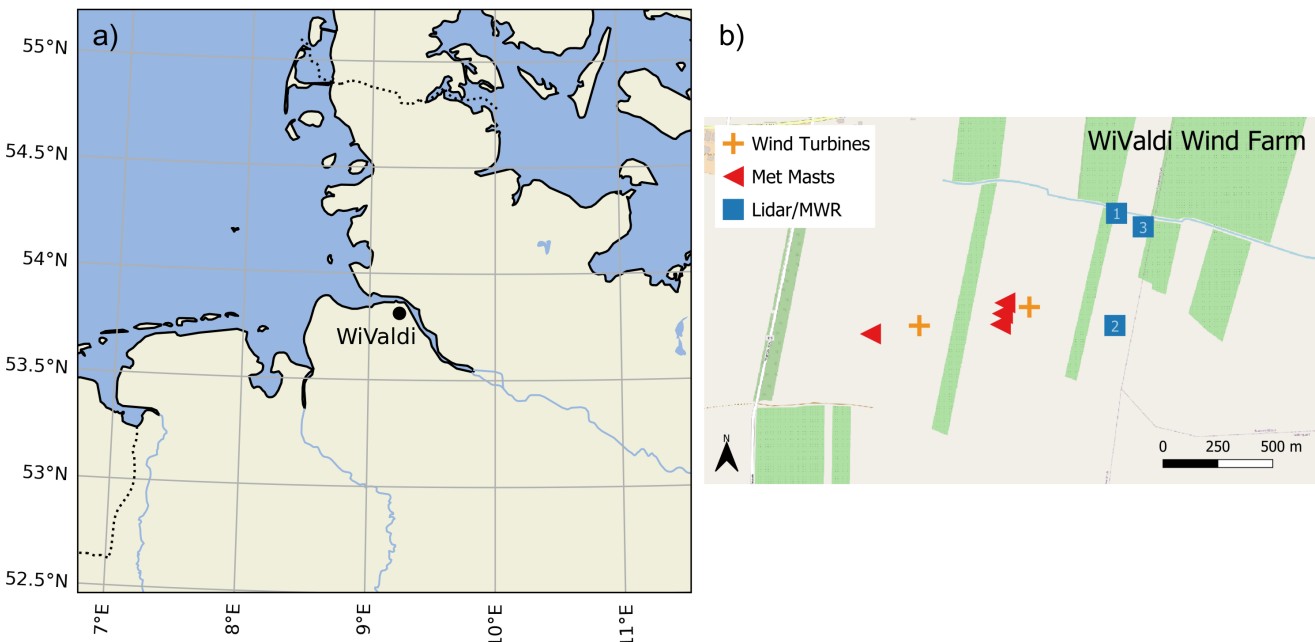

**Figure 1.** (a) Map of northern Germany showing the location of the WiValdi wind farm. Dashed lines indicate country borders. (b) Detailed layout of WiValdi, indicating the locations of the wind turbines, meteorological masts, MWR, lidar, and underlying simple terrain. The blue squares represent the location of both the MWR and lidar, with the overlaid numbers indicating their different positions in chronological order: November 2020 - March 2022 (1), October 2022 - May 2024 (2), and May 2024 - present (3). Background map: ©OpenStreetMap contributors 2025. Distributed under the Open Data Commons Open Database License (ODbL) v1.0.

wind variations associated with convective cold pools, but we limit interpolation to within surrounding reliable measurements (signal-to-noise ratio from 0 to -25) and do not interpolate across more than 1 missing value.

Adjacent to the lidar, there is a HATPRO G5 microwave radiometer (MWR; Rose et al., 2005) manufactured by Radiome-
ter Physics GmbH that was installed on 26 November 2020 and which provides vertical profiles of temperature and humidity. With an associated ground weather station for obtaining surface environmental conditions, an MWR retrieves vertical profiles through usage of multiple brightness temperature measurements within the oxygen and water vapor absorption bands. Absolute calibration using liquid nitrogen is performed about every half year, with a gain calibration using the internal hot target being carried out every 30 minutes and noise calibration is automatically performed internally with high frequency during normal
operations.

The HATPRO G5 MWR can perform zenith and boundary-layer scans to obtain temperature profiles, while humidity profiles are determined only from the zenith scans. Zenith scans point directly upwards and retrieve measurements up to 10 km altitude, with a general decrease in spatial resolution with increasing height (especially above 2 km). Boundary-layer scans are performed at 10 different elevation angles ranging from 4-90° up to 1.2 km altitude towards the south (185° azimuth), which





can more accurately capture vertical gradients in temperature and humidity than zenith scans. The MWR at WiValdi performs a boundary-layer scan lasting about 100 seconds every ten minutes. Otherwise, the instrument measures at zenith (90° elevation) to obtain 1-minute temperature and humidity. To better capture rapid environmental variations associated with convective cold pool passages, we will use the 1-minute zenith vertical profile data for this study.

## 2.2 Cold pool detection algorithm

Following guidance from recent European cold pool studies (Kirsch *et al.*, 2021; Kruse *et al.*, 2022), we identify convective cold pools at WiValdi using 1-minute time series data from the 2-m ground weather station attached to the MWR. High-frequency observations on the minute-scale are crucial for capturing the rapid environmental changes associated with cold pools, while near-surface observations ensure identification of the cold pool thermodynamical signal which is strongest near the surface (e.g., Barnes and Garstang, 1982; Kirsch *et al.*, 2021). Using 2-m in-situ measurements across 4 years (November 150 2020 - November 2024), we detect cold pool passages throughout the year using the following procedure:

- Smooth the 2-m $\theta_v$ time series using 11-minute rolling averages, following Kruse *et al.*(2022) which extends the methodology of de Szoeke *et al.*(2017). This smoothing reduces the effect that random $\theta_v$ fluctuations could have on misidentifying the starting time of a cold pool passage, while $\theta_v$ rather than temperature is used to account for pressure and moisture variations.

- Isolate time periods of *continuous* 2-m $\theta_v$ decreases to further minimize misidentification caused by random $\theta_v$ fluctuations, and which follows from observed cold pool horizontal structure where temperature decreases approximately linearly from the edge to the center (Kirsch *et al.*, 2024).

- Identify the time step when the smoothed 2-m $\theta_v$ continuously decreases by at least 0.5 K as '$T_0$', following Kirsch *et al.*(2021).

- Continuous-$\theta_v$-decrease time periods include at least one time step of measurable rainfall exceeding 1 mm hr$^{-1}$ and a positive daily wind anomaly within +/- 10 minutes of $T_0$. This rainfall threshold is used to remove instances of very weak convection or possible rainfall measurement error, while the wind anomaly is inspired by Kruse *et al.*(2022) and verifies a more significant cold pool gust front strength (given our interest in quantifying cold pool impacts on wind turbines).

- There is no rainfall from $T_0$-30 to $T_0$-10 minutes to reduce environmental contamination by recent or nearby convection, as well as to reduce false positive cases that could actually be related to cold fronts with along-frontal precipitation.

- $\theta_v$ continuously decreases by at least 1.5 K. This $\theta_v$ threshold is chosen to match the temperature decrease threshold of Kruse *et al.*(2022), with their study in the Netherlands being similarly situated in coastal flatlands. We have increased the threshold to -1.5 K from the -2 K used in Kirsch *et al.*(2021) due to WiValdi's coastal location and greater environmental moisture, which would likely reduce the amount of evaporative cooling.





- A $\theta_v$ drop of 1.5 K occurs within 30 minutes of $T_0$. This $\theta_v$ gradient threshold ensures a more robust cold pool passage, while minimizing occurrences of more-gradual cooling associated with phenomena outside the scope of this study (i.e., sea breezes, diurnal cycle, etc.). The 30-minute time constraint falls within the range of past work (20 minutes for Kruse *et al.*, 2022; 60 minutes for Kirsch *et al.*, 2021). For context, changing this time constraint to 20 minutes or 60 minutes would decrease or increase detected cases by one-third, respectively.

- Finally, we prescribe that $\theta_v$ must recover at least somewhat within 60 minutes of $T_0$ from its minimum value. This aligns with known cold pool horizontal structure (Kirsch *et al.*, 2024) and further reduces false positives associated with cold frontal passages. The inclusion of this criterion only removes a few cases from our dataset, but visual inspection reveals that these removed cases resemble cold fronts (not shown).

Cold pool events are included in our dataset if they meet the above criterion. We conservatively define the 'pre-cold
pool environment' as $T_0$-30 minutes, with findings by Kirsch *et al.*(2021) and Kruse *et al.*(2022) indicating that near-surface environmental conditions do not significantly change until at least $T_0$-15 minutes for cold pool passages. As we require rainfall to be detected at WiValdi for a cold pool to be counted, we exclude any cold pool passages where the parent convective cell's rainfall misses WiValdi but the cold pool periphery crosses over the wind park.

## 2.3 Convective cell tracking

While weather station and meteorological mast observations can and have been used to identify convective cold pools (e.g., Kirsch *et al.*, 2021; Hoeller *et al.*, 2024), additional radar measurements can be useful to (1) confirm the presence of a parent convective cell linked to the ground-based cold pool and (2) provide comparison between radar-derived convection characteristics and near-surface cold pool characteristics. Therefore, to supplement the WiValdi observational network, we track convective cells associated with the detected cold pool cases using DWD radar composite datasets.

We utilize the radar reflectivity (WN) composite dataset from DWD, which has 1 km x 1 km horizontal resolution and 5-minute temporal resolution over Germany. This dataset is derived from 5-minute terrain-following scans of the German C-band radar network with a range resolution of 1 km and azimuth resolution of 1° (Wapler, 2021; Wilhelm *et al.*, 2023). We then employ the Tracking and Analysis of Thunderstorms (TATHU) toolset (Uba *et al.*, 2022; Sena *et al.*, 2024) to systematically detect and track convective systems within the WN dataset. With TATHU, convective elements, or objects, are identified using
a chosen threshold from a given 2D field from which a corresponding bounded polygon is drawn. Given the polygon boundary at each time step and a user-provided polygon overlap area percentage, identified convective elements can be linked in time, including whether storms undergo splitting or merging during their lifecycle. Convective element characteristics (weighted centroid position, shape, size) are provided at each time step and statistical quantities of the given 2D field related to each convective element can be obtained.

For this study, we define a convective cell as a contiguous area of at least 15 km$^2$ with a minimum radar reflectivity of 19 dBZ, with a 10% polygon overlap criterion used to connect convection snapshots in time. While the size threshold is identical to the DWD detection and nowcasting tool KONRAD, the reflectivity threshold of 19 dBZ is much lower than





the 46 dBZ chosen by recent studies (Wapler, 2021; Wilhelm *et al.*, 2023). Goudenhoofdt and Delobbe (2013) proposed a minimum 40-dBZ threshold to identify convective precipitation over Belgium, suggesting that lower thresholds could lead to

a higher potential for storms to be classified as splitting or merging due to closer proximity with nearby convection. However, to ensure identification of a precipitating convective cell close in time to the observed cold pool (which is detected without any convection-type rainfall classification), as well as to more fully capture a convective cell's lifecycle from early development to late decay, we employ this lower 19-dBZ reflectivity threshold for cell tracking.

## 3    Temporal Evolution of Cold Pool Characteristics

Using the detection criterion outlined in Section 2.2, we identify 120 convective cold pools impacting WiValdi from 2020-

|  | Cases | Data Availability | 2-m Median Pre-Event Environment [$T_0$-30] | | |
|---|---|---|---|---|---|
|  |  |  | $\Theta_v$ [ ˚C] | $q_{sat} - q$ [g kg$^{-1}$] | U [m s$^{-1}$] |
| **Jan** | 5 | 86% | 5.98 | 0.91 | 6.86 |
| **Feb** | 7 | 98% | 5.48 | 0.93 | 4.26 |
| **Mar** | 11 | 72% | 4.07 | 1.04 | 3.76 |
| **Apr** | 13 | 57% | 5.83 | 1.33 | 3.07 |
| **May** | 11 | 52% | 12.57 | 2.15 | 2.88 |
| **Jun** | 11 | 44% | 16.95 | 2.72 | 4.87 |
| **Jul** | 18 | 65% | 19.80 | 4.02 | 2.25 |
| **Aug** | 17 | 72% | 19.89 | 3.16 | 2.99 |
| **Sep** | 3 | 68% | 12.74 | 1.83 | 2.90 |
| **Oct** | 12 | 84% | 12.14 | 2.12 | 3.47 |
| **Nov** | 7 | 100% | 7.89 | 1.16 | 3.86 |
| **Dec** | 5 | 97% | 6.74 | 1.02 | 6.85 |

**Table 1.** Monthly breakdown of detected cold pool cases, 1-minute ground station data availability, and 2-m median pre-event environment ($T_0$-30 minutes) in terms of $\theta_v$ (˚C), saturation deficit ($q_{sat}$ - q, g kg$^{-1}$), and total wind speed (m s$^{-1}$). Gray shading indicates the warm season at WiValdi as shown by $\theta_v$.


2024. Table 1 highlights the non-uniform distribution of cold pool events throughout the year, with generally more cases in the warmer months of May through October. However, the lower availability of 1-minute ground station observations in warmer months indicates a likely under-count of cold pool passages in total. The greater pre-event surface saturation deficit ($q_{sat}$ - q) during May-October indicates a likelihood for greater evaporative cooling of rainfall during warmer months, though WiValdi

generally has a lower saturation deficit during summer months than the 4.7 g kg$^{-1}$ median saturation deficit found by Kirsch



*et al.*. (2021) during summer months farther inland near Hamburg. With the near-surface temperature perturbation found to have a relatively strong inverse relationship with pre-event saturation deficit (Kirsch *et al.*, 2021), the reduced saturation deficit at WiValdi suggests that WiValdi cold pools would likely be weaker on average (in terms of surface temperature decrease) than those observed near Hamburg. Higher background wind speeds are generally found in winter months in agreement with long-term wind climatologies over Europe (e.g., Molina *et al.*, 2021). As with the variability of pre-event environments, there is a variety of parent convective cells associated with the detected cold pools, which are represented by the radar reflectivity snapshots in Figure 2 from the nearest radar times to their respective $T_0$ times.

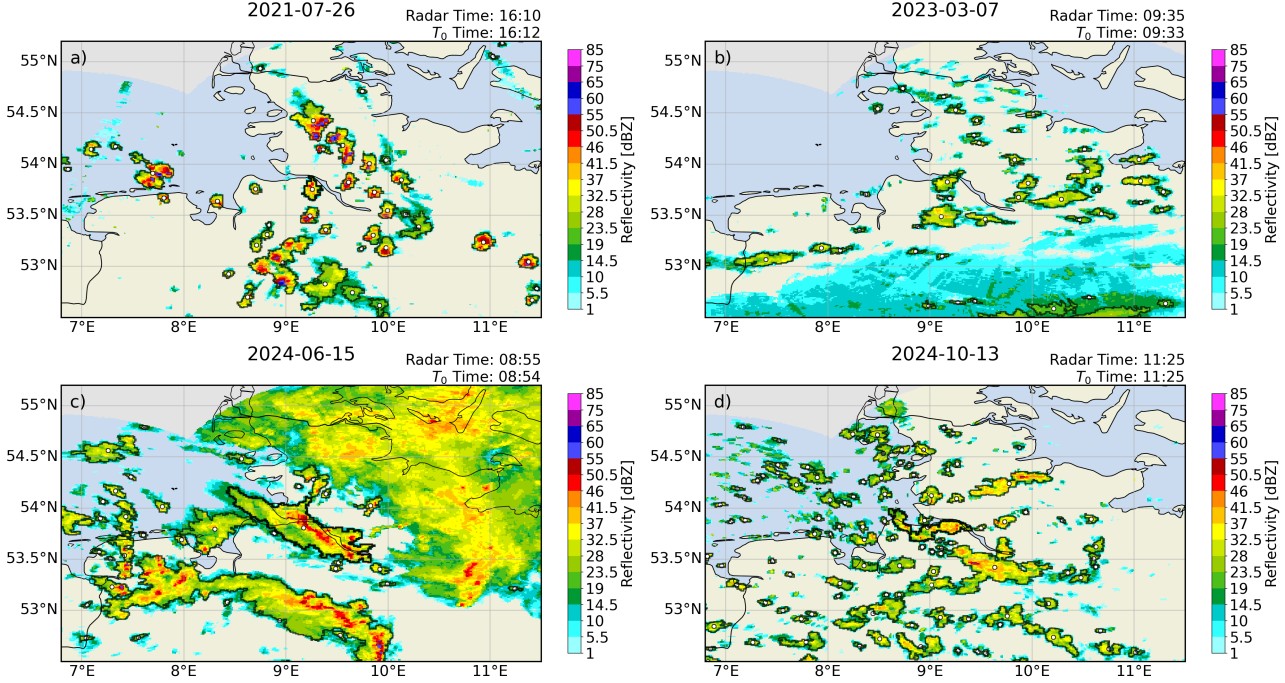

**Figure 2.** Snapshots of convection situations throughout northern Germany in terms of radar reflectivity (dBZ, colors) for four parent convective cells representing the convection variability associated with the detected cold pool cases. Magenta dot at center indicates the location of WiValdi, with white dots identifying the polygon centroids for tracked cells. Black polygon outlines reflect when a polygon overlaps with WiValdi (a, c, d), otherwise polygon outlines are blue.

To gain an understanding of the bulk surface characteristics associated with the detected cold pools at WiValdi, we first examine changes in 2-m ground station observations relative to $T_0$ (Fig. 3). The median wind speed increases (dark blue) and the wind direction shifts more westerly (light blue) starting around $T_0$-15 minutes, with a more rapid increase in wind speed after $T_0$-5 minutes. This closely follows the temporal evolution of wind speed observed by Kirsch *et al.*. (2021), with a relative increase of 1-1.5 m s$^{-1}$ by $T_0$+5 minutes being similar to their 2 m relative increase of 1.5 m s$^{-1}$. The median surface wind direction shifts towards westerly by up to 40° during the detected events. Using $T_0$-30 minutes as a proxy for the pre-event environment, we find a median $\theta_v$ relative decrease of 2.5 K (red) during the cold pool passages that occurs in



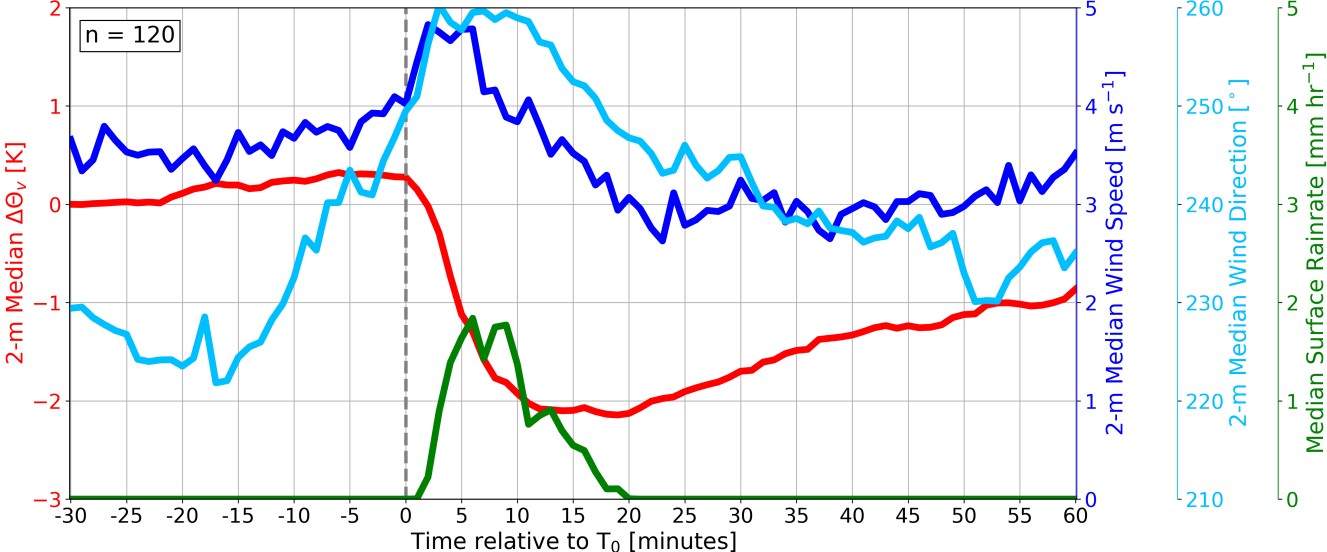

**Figure 3.** Median surface characteristics for the detected cold pool cases as measured by the 2-m ground station attached to the MWR relative to $T_0$ (dashed gray line), in terms of $\Delta\theta_v$ from $T_0$-30 minutes (K; red), wind speed (m s$^{-1}$; dark blue), wind direction (°; light blue), and rainfall (mm hr$^{-1}$; green). Detected cold pool sample size is given in the upper left.

the span of $\sim$20 minutes, starting at $T_0$-5 minutes and reaching a minimum around $T_0$+15 to $T_0$+20 minutes. As in Kirsch *et al.*. (2021), the $\theta_v$ drop onset slightly precedes $T_0$ by a few minutes due to our chosen detection thresholds. Shortly after this onset, measurable rainfall is observed which peaks at 2 mm hr$^{-1}$ and lasts $\sim$20 minutes. With a surface wind speed increase, wind direction shift, $\theta_v$ decrease, and measurable rainfall, the quintessential convective cold pool characteristics are observed at WiValdi.

After the cold pool gust front passage, which is strongly indicated by the rapid wind speed increase and $\theta_v$ drop onset at $T_0$-5 minutes, the wind speed gradually weakens to below the pre-event value and the wind direction gradually shifts back towards $\sim$230° as the cold pool interior and parent convective cell pass by. The surface $\theta_v$ similarly recovers towards its pre-event state, though notably remains 1 K colder by $T_0$+60 minutes than at $T_0$-30. This may suggest that cold pools are still propagating past WiValdi even 1 hour after the leading gust front, or at minimum that there is a lasting effect from the observed

cold pools on the near-surface environment of at least 1 hour. Kruse *et al.*(2022) noted that temperature recovery typically occurred only after 2 hours for their cases in the Netherlands. So, while the kinematic changes accompanying the gust front are rapid (5-10 minutes) and would thus likely have an impact on turbine structural loads, the thermodynamic environment more gradually changes, with longer-lasting implications for turbine wakes and power generation.

        Moving above the surface to turbine hub-height, we want to quantify how much the kinematic and thermodynamic envi-

ronment changes from the pre-event state. For wind speed and direction (Fig. 4a-b), we perform comparisons between 100 m lidar winds and an 85 m ultrasonic anemometer on the inflow meteorological mast, noting a decreased sample size of de-





tected cold pool events from those observed by the ground station (Fig. 3). Both measurement systems similarly find a relative wind speed increase of 4 m s$^{-1}$ on top of the background flow that peaks around $T_0$+5 minutes (Fig. 4a), with recovery to the pre-event wind speed typically by $T_0$+20 minutes. The 10th and 90th percentiles show a similar spread of wind speed changes, indicating that the 1-minute lidar scans are able to reasonably capture both the median and range of observed cold pool-associated hub-height wind speed variations. Of note, the timing of the gust front and recovery to the background flow as captured by the sonic tend to slightly precede that of the lidar by a couple minutes, which can largely be explained by the ∼1-km instrument separation (with the more-westward sonic likely being impacted by cold pool passages before the lidar in most instances). Wind direction comparisons show a similar agreement between lidar and sonic (Fig. 4b). There is a peak wind direction change of +15° between $T_0$ and $T_0$+10 minutes which then gradually decreases, though the median relative wind direction change for both instruments does not recover to its pre-event value even after 1 hour. Though there is a sample size difference between sonic and lidar, the relative magnitude and spread of kinematic variations during cold pool passages at WiValdi are remarkably consistent between the *in-situ* and remote-sensing observations.

As past cold pool studies have noted an increase in turbulence associated with the gust front (Droegemeier and Wilhelmson, 1987; Lombardo *et al.*, 2014; Canepa *et al.*, 2020), and since turbulence strength correlates with stability regime and impacts turbine structural stress (e.g., Abkar and Porte-Agel, 2014, Englberger and Doernbrack, 2018), assessing short-term variations in turbulence is relevant for both convective cold pool and wind energy applications. The TKE dissipation rate is a metric for quantifying the turbulence strength, and is estimated herein every 1 minute using high-frequency (20 Hz) observations from the 85-m ultrasonic anemometer. Following Bodini *et al.*, (2019), the theoretical turbulence model of Kolmogorov (Kolmogorov, 1941) is fit to the second-order structure function calculated from the sonic data using a range of time lags $\tau$ between 0.15 s and 1.05 s. Data inspection confirms that the sonic measurements within this time interval closely follow the Kolmogorov slope (k = -5/3), and hence lie within the inertial subrangeof the turbulence energy spectrum. During the detected cold pool passages, there is a relative increase in median $\varepsilon$ from the pre-event environment up to $10^{-2}$ around $T_0$, indicating an increase in turbulence strength associated with the cold pool gust front (Fig. 4c). As turbulence tends to be greater during daytime hours and summertime months (Bodini *et al.*, 2019), which is also when most thunderstorms and thus convective cold pools would occur, the positive change in $\varepsilon$ during the gust front would be in addition to this greater background turbulence pattern. After $T_0$, there is a decrease in $\varepsilon$ coinciding with the subsidence in wind speeds (Fig. 4a) and cooling of the near-surface atmosphere (Fig. 4d).

In contrast to the kinematic variations (Figs. 4a-b), the hub-height thermodynamic variations have noticeable disagreements in terms of $\theta_v$ between 1-minute mast instrumentation and 1-minute MWR zenith retrievals at 85 m (Fig. 4d). While the median and 10th-90th percentile spread of relative $\theta_v$ changes follow a similar trend until the $\theta_v$ drop onset, a sizable proportion of MWR profiles remain warmer than the pre-event environment after $T_0$ in contrast to the in-situ observations. The 85 m MWR measurements are able to capture the first minutes of cooling after the gust front passage, but then the median values from the mast and MWR deviate from each other starting around $T_0$+5 minutes after a relative cooling to -0.5 K. The median mast observations indicate additional cooling to nearly -2 K by $T_0$+20 minutes, while the MWR profiles remain largely constant. As surface rainfall rates also typically increase within this time period from $T_0$+5 to $T_0$+20 minutes (Fig. 3), it is

**Figure 4.** Hub-height variations relative to the pre-event environment ($T_0$-30 minutes) and relative to $T_0$ (dashed gray line) for (a) wind speed (m s$^{-1}$; dark blue), (b) wind direction (°; light blue), (c) TKE dissipation rate $\varepsilon$ (m$^2$ s$^{-3}$; orange), and (d) $\theta_v$ (K; red). Solid colored lines indicate 100-m lidar (a-b), (c) 85-m sonic $\varepsilon$, and (d) 85-m MWR, while dashed colored lines in (a-b, d) indicate 85-m in-situ meteorological mast observations. Solid brown line in (c) is the 11-minute rolling mean $\varepsilon$. Shading and thinner colored dashed lines indicate the 10th-90th percentile spread. Detected cold pool sample sizes given in the upper right for each instrument.

likely that raindrops on the MWR radome are interfering to some extent with the zenith retrievals for the measured frequencies in the microwave spectrum, particularly those in the water vapor absorption band. Foth *et al.*. (2024) note that MWR retrieval algorithms generally decrease in accuracy for greater rainfall rates, with the zenith elevation angle having the largest positive brightness temperature biases (>3 K) across most measured frequencies compared with lower elevation angles. Additionally, as the thermodynamic vertical gradient between the surface and turbine hub-height also increases between $T_0$ and $T_0$+20 minutes



(see Fig. 7), and since MWR zenith scans are less capable at capturing large vertical gradients, the MWR would be expected to have less accuracy during the time period of greatest cooling even if rainfall was not occurring. Both measurement systems show some recovery towards the pre-event state after $T_0$+20 minutes, though the MWR observations for most detected cold

pool cases tend to have a persistent warm bias through $T_0$+60 minutes. Overall, the meteorological mast shows good agreement with the temporal evolution of $\theta_v$ given by the surface station (Fig. 3), while the MWR profile measurements tend to underestimate the observed cooling within the cold pool interior. As such, we exclude the MWR profiles from our subsequent analyses.

To better contextualize the above temporal evolution findings, we provide a brief case study for a cold pool that was

detected on 15 June 2024 at 0854 UTC that is at the stronger end of the range of detected cold pool events. This cold pool occurred on a convectively-active day and was associated with a mesoscale convective system in northern Germany (Fig. 2c). Stratiform rainfall preceded the cold pool passage, and additional cold pools were detected at WiValdi later that morning (not shown). As the gust front approached, there was an increase in wind speeds from 7 to 14 m s$^{-1}$ (Fig. 5a) alongside a wind direction shift from 195 to 235° (Fig. 5b) over the course of a few minutes. A significant increase in $\varepsilon$ from $T_0$-30 minutes to

$T_0$ highlights the greater turbulence strength associated with this gust front passage relative to the background flow (Fig. 5c). Immediately after the gust front, there was a relative drop in $\theta_v$ of almost 3 K at the surface and 2.5 K at hub height over the course of 10-15 minutes, alongside intense rainfall up to 18 mm hr$^{-1}$ which lasted until $T_0$+25 minutes (Fig. 5d). Wind speeds quickly subsided with the near-surface stabilization, while wind direction and $\varepsilon$ more gradually shifted back towards their pre-event values. After 1 hour, the kinematic and thermodynamic near-surface atmosphere remained altered from the pre-event

environment, though changes beyond about $T_0$+30 minutes are likely also impacted by subsequent convection.

## 4    Vertical Structure of Cold Pool Characteristics

As the wind speeds induced by cold pools often maximize above the surface within the height range of wind turbines, we isolate the vertical structure associated with the peak gust front strength ($T_0$ to $T_0$+5 minutes) as shown in Figures 3 and

4 using averaged vertical profiles up to 1 km height. Vertical profiles of relative wind speed from the lidar (Fig. 6a) highlight the increased winds associated with the cold pool gust front, with the largest relative increases occurring at 100 m. The median wind speed shows a relative increase from the pre-cold pool environment up to about 650 m, which provides a rough estimate for the median gust front depth at WiValdi. The relative wind direction profiles indicate a maximum shift of +15° (e.g., more westerly on average) close to the surface in most cases (Fig. 6b), though changes within the rotor layer still typically exceed

+10°. With the median relative wind direction showing a positive wind direction shift up to 700 m, this provides additional support for an approximate cold pool gust front depth of between 650-700 m. Notably, Kirsch *et al.*(2021) found an estimated median cold pool depth of 750 m near Hamburg using perturbation pressure extrapolation of mast data, which closely aligns with the zero-crossing height of the kinematic relative vertical profiles using the lidar at WiValdi.





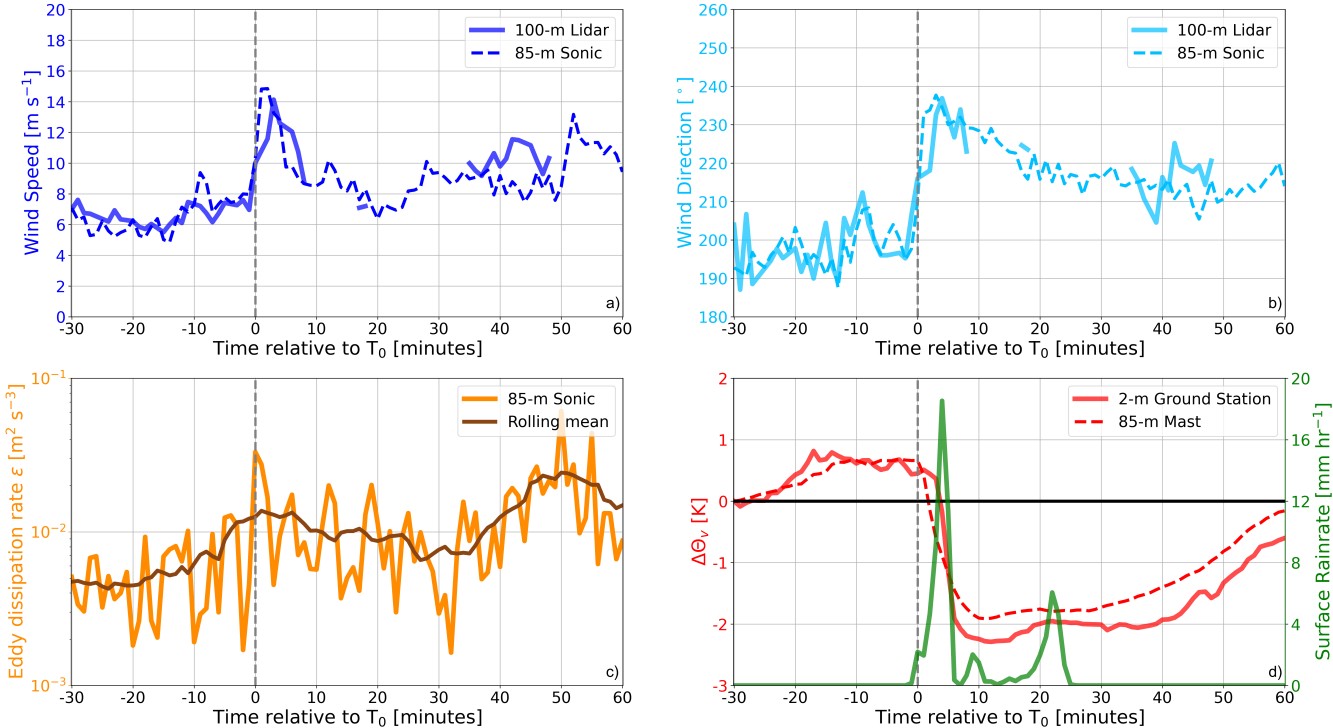

**Figure 5.** Case study on 15 June 2024 for $T_0$ = 0854 UTC. (a) wind speed (m s$^{-1}$; dark blue), (b) wind direction (°; light blue), (c) TKE dissipation rate $\varepsilon$ (m$^2$ s$^{-3}$; orange), and (d) $\Delta\theta_v$ (K; red) and rainfall (mm hr$^{-1}$; green). Solid colored lines indicate 100-m lidar (a-b), 85-m sonic (c), and 2-m ground station (d) data. Solid brown line in (c) is the 11-minute rolling mean $\varepsilon$. Dashed colored lines indicate 85-m sonic (a-b) and 85-m mast (c) observations. Lidar data with a signal-to-noise ratio below -25 has been excluded from (a) and (b).

Focusing on the height region of typical turbines, we want to obtain a more-detailed understanding of the vertical profile variations associated with the detected convective cold pools. There is a more-limited sample size when all mast instrumentation heights are available, but the mast provides sufficient context for comparing with the remote-sensing vertical profiles. A typical gust front peaks in strength around the turbine hub-height at +3 m s$^{-1}$, exhibiting a nose shape within the turbine rotor layer (34-150 m; dashed black lines) as has been observed by past work (Lombardo *et al.*, 2014; Gunter and Schroeder, 2015; Canepa *et al.*, 2020), with the vast majority of cases showing increased wind speeds from the background flow (Fig. 7a). While it appears that some gust fronts do not cause an increase in wind speed, this is actually a result of kinematic variations between $T_0$-30 and $T_0$ (see 10th percentile in Fig. 4a) which complicates the quantification of localized relative wind speed changes during the short time period of the gust front passage. Nevertheless, there is good agreement in the vertical shape and spread of wind speed above hub-height for the mast and lidar, further validating the remotely-sensed wind speeds. Notably, the mast data captures a greater wind speed change with height over the lower half of the rotor layer than the upper half. For the relative wind direction (Fig. 7b), a peak of +15° occurs close to the surface, in contrast to the wind speed peak that occurs near hub-height. The lidar relative wind direction changes between about 100-150 m are roughly comparable to the mast observations





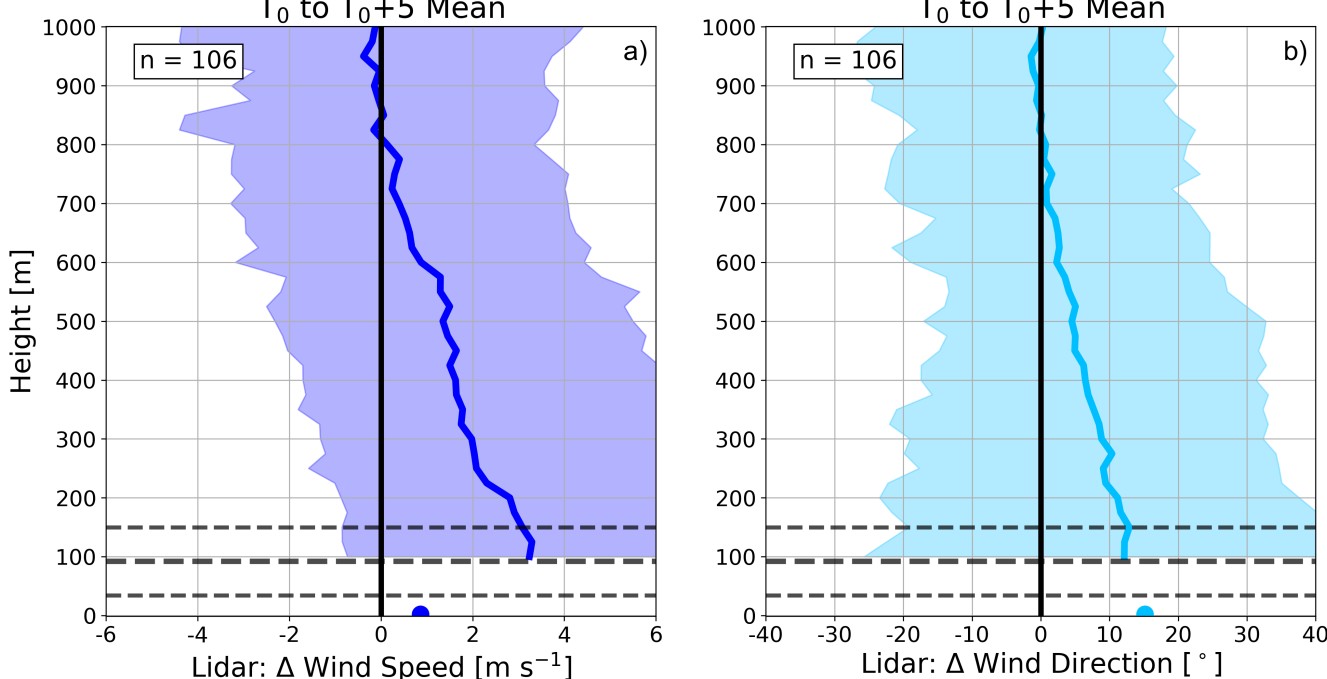

**Figure 6.** Mean vertical profiles over the given time periods relative to the pre-event environment ($T_0$-30 minutes) for (a) wind speed (m s$^{-1}$; dark blue) and (b) wind direction (°; light blue). Dots correspond to the median 2-m ground station observations averaged over the given time periods. Solid colored lines indicate the median conditions, while shading indicates the 10th-90th percentile spread. Horizontal dashed black lines indicate the turbine bottom (34 m), hub (92 m), and top (150 m). Detected cold pool sample size given in the upper left.

between these heights, qualitatively indicating that the lidar reasonably well captures the maximum wind direction changes accompanying the gust front. While there is a slightly greater wind direction change over the lower half of the rotor layer similar to the wind speed, the difference with the upper half is not quite as significant. The 'nose' in $\theta_v$ (Fig. 7c), with a lesser

relative change at hub height compared to above and below, corresponds to the nose-like shape found in the wind speeds and is likely a manifestation of the gust front vertical circulation being present over WiValdi during this time period.

By the time of maximum cold pool cooling ($T_0$+15 to $T_0$+20 minutes), the wind speeds have largely relaxed back towards the pre-event flow below hub height, though they remain increased by 1 m s$^{-1}$ at turbine top (Fig. 7d). The wind direction adjustment is comparatively more subtle with some reduction in direction shift (Fig. 7e), but still being 5-10° more westerly on

average from $T_0$-30 minutes throughout the rotor layer, indicating that the wind direction typically takes longer to recover than the wind speed after cold pool passages. Since the gust front time period (Fig. 7c), the relative $\theta_v$ shows greater cooling at all mast heights (Fig. 7f), being greatest near the surface at -2 to -2.5 K in agreement with the known vertical structure of the cold pool thermodynamical signal (e.g., Barnes and Garstang, 1982; Kirsch *et al.*, 2021). The 1-minute ground station observations seem to adequately capture the short-term environmental variations associated with the cold pools (with the possible exception



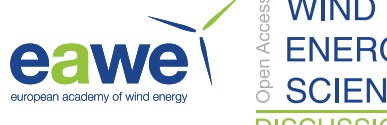

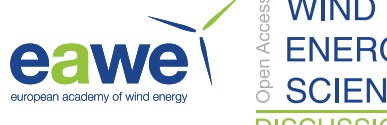

**Figure 7.** Mean vertical profiles within the turbine rotor layer over the given time periods for (a, d) wind speed (m s$^{-1}$; dark blue), (b, e) wind direction (°; light blue), and (c, f) $\theta_v$ (K; red) changes relative to $T_0$-30 minutes using meteorological mast (solid) and lidar (a-b, d-e; dashed) measurements. Horizontal dashed black lines indicate the turbine bottom (34 m), hub (92 m), and top (150 m). In-situ ground station measurements are shown as dots at 2 m. Median values shown in bold and thicker dashed colored lines, with 10th and 90th percentiles shown in shading and smaller dotted colored lines. Detected cold pool sample sizes per instrument given in the upper right.

of the initial $\theta_v$ drop relative to the mast data; Fig. 7c). The observed rotor-layer cooling suggests stabilization throughout the profile within the cold pool interior, which will be explicitly quantified in the following section.





## 5  Wind Turbine Rotor Layer

Now that we have examined how the temporal evolution and vertical structure of the background kinematic and thermo-dynamic environment is affected by convective cold pools at WiValdi, we next investigate variations in quantities that affect
power output and structural loads across the rotor layer associated with cold pool passages. As indicated by the wind speed nose near hub height around $T_0$ to $T_0$+5 minutes (Fig. 7a), there is increased lower rotor-layer shear from the pre-event environment of 1.5 m s$^{-1}$ (over a depth of 51 m) associated with the gust front alongside comparatively negligible changes in upper rotor-layer shear (Fig. 8a). Therefore, the total rotor-layer shear changes (over a depth of 115 m) induced by the cold pool gust front are almost exclusively encapsulated by shear changes below hub height. Upper rotor-layer shear increases more
thereafter, peaking at 1 m s$^{-1}$ (over a depth of 64 m) about 30 minutes after the gust front. The total rotor-layer shear remains increased by at least 1 m s$^{-1}$ even 1 hour after $T_0$, indicating that relative increases in shear-induced loads on the turbines would likely persist well after the cold pool leading edge. Similar to wind shear, the wind veer shifts the greatest around the time period of the gust front (Fig. 8b), noting that the relative wind veer is negative during WiValdi cold pool passages owing to greater wind direction changes closer to the surface (Fig. 7b). Veer across the lower part of the turbine rotor peaks at -6°
around $T_0$, with the total rotor-layer veer reaching -8° and the upper-layer veer peaking a few minutes later at -4°. Overall, the largest change in rotor-layer veer is associated with the cold pool leading edge, but a median shift in rotor-layer veer of -4° remains even after 1 hour. Given the nose in relative $\theta_v$ around turbine hub height (Fig. 7c), there is a relative increase in static stability of 14 K km$^{-1}$ over the lower rotor-layer and relative decrease in stability (greater instability) of 7 K km$^{-1}$ over the upper rotor-layer associated with the cold pool gust front (Fig. 8c). This asymmetric change in stability across the
rotor layer continues until $T_0$+15 minutes, but it is unclear how the turbine wake would behave in this complex stability regime environment. Over the course of a thunderstorm passage, there is a general trend towards increasing stability compared with the pre-event environment as the near-surface cooling persists, remaining 7 K km$^{-1}$ more stable (0.8 K across the turbine rotor) after 1 hour. This longer-term stabilization throughout the rotor layer would tend to lengthen turbine wakes in the stream-wise direction (Abkar and Porte-Agel, 2014), increasing the potential for downstream turbines to experience velocity deficits,
decreased turbulence, and produce less power.

Beyond shear and veer variations, we also seek to quantify the change in power generation that would be expected for a stand-alone turbine impacted by a cold pool. This has been modeled using a simple piecewise wind-power relationship (Wildmann *et al.*, 2022), wherein the cut-in, rated, and cut-outhub-height wind speeds are provided alongside the maximum turbine power output to obtain the 'estimated' power generation. Estimated power is taken to linearly increase between the cut-
in and rated wind speeds (i.e., partial load range), and we do not consider impacts from turbine operational adjustments (e.g., pitching, yawing) on the power output. For the 4.2-MW WiValdi wind turbines, we have a 2.5 m s$^{-1}$ cut-in, 13.1 m s$^{-1}$ rated, and 25 m s$^{-1}$ cut-out wind speed. With the 100 m lidar wind speeds being used as the hub-height wind input for this power estimate calculation, it is important to first determine where the background winds lie on the power curve to understand how much the subsequent cold pool wind speed perturbations may cause fluctuations in power generation. The 100-m pre-event
environment typically has wind speeds of 9 m s$^{-1}$, with the 10-90th percentiles ranging from 4-15 m s$^{-1}$ (Fig. 9a). Therefore,



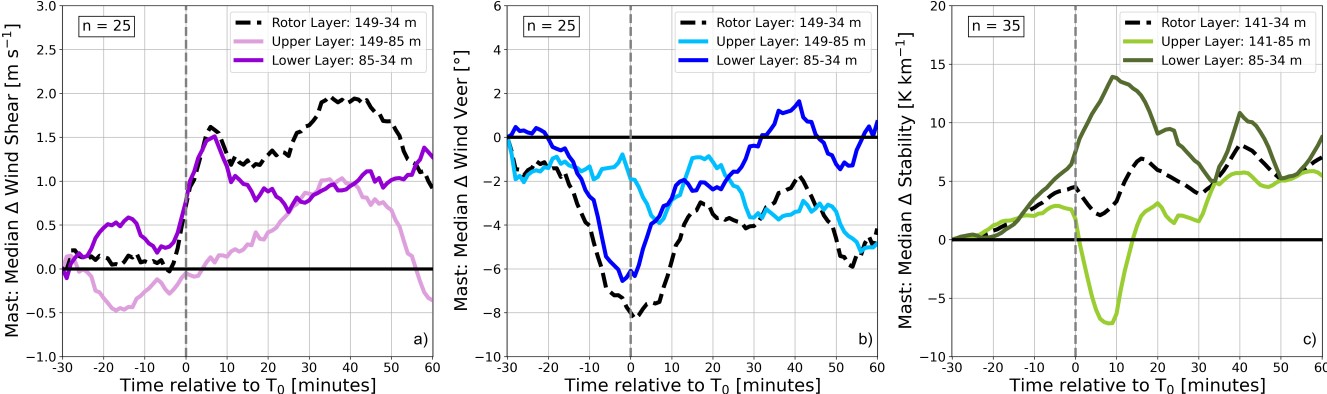

**Figure 8.** Median changes in meteorological mast (a) wind shear (m s$^{-1}$), (b) wind veer (°), and (c) static stability $\Delta\theta_v/\Delta z$ (K km$^{-1}$) relative to the pre-event environment ($T_0$-30 minutes) over the lower layer (85-34 m; darker lines), upper layer (149-85 m; lighter lines), and full rotor layer (149-34 m; dashed). Sample sizes given in the upper left.

background hub-height winds are generally above the cut-in wind speed and within the partial load part of the power curve, indicating that most detected cold pool gust fronts would likely cause an increase in power output. When considering the climatological wind speeds at WiValdi (dashed lines, Fig. 9a), the pre-event flow is already elevated, suggesting that convective conditions before the cold pool time periods are already associated with a potential for greater than average power output. Lidar

median wind speeds rise by 4 m s$^{-1}$ in addition to the background flow during gust front passages (Fig. 4a), which reaches the rated wind speed, and gust front amplitudes range from 3-6 m s$^{-1}$. As noted by Kelly *et al.*(2021), wind ramps starting below the rated wind speed and ending above the rated wind speed significantly contribute towards increased turbine structural loads, which aligns with the typical WiValdi cold pool. Using the above-mentioned power estimation and 100 m winds, we estimate a median increase in power output up to 50% around $T_0$ to $T_0$+5 minutes associated with the gust front (Fig. 9b).

Estimated power increases can exceed 100% (for cases starting from weak background flow), and power generation remains larger than the pre-event environment for about 30 minutes. We have a limited sample size of observed power output during the detected cold pools, but power observations from the westernmost WiValdi turbine during summertime months in 2024 broadly agree with having a significant peak associated with the gust front and increased power lasting about 30 minutes. After the gust front maximum, there is some indication of a relative reduction in observed wind power compared with the pre-event

state, likely stemming from a relative decrease in hub-height wind speeds alongside the near-surface atmosphere stabilization. As the observed power changes are complicated by continued turbine testing during the available time period and since the turbine was not always allowed to produce up to the rated power output, a larger observed power climatology in the future is needed to more reliably contextualize the estimated power generation. Nevertheless, we determine that noteworthy wind power increases would be expected due to the majority of cold pool gust fronts in the vicinity of WiValdi.





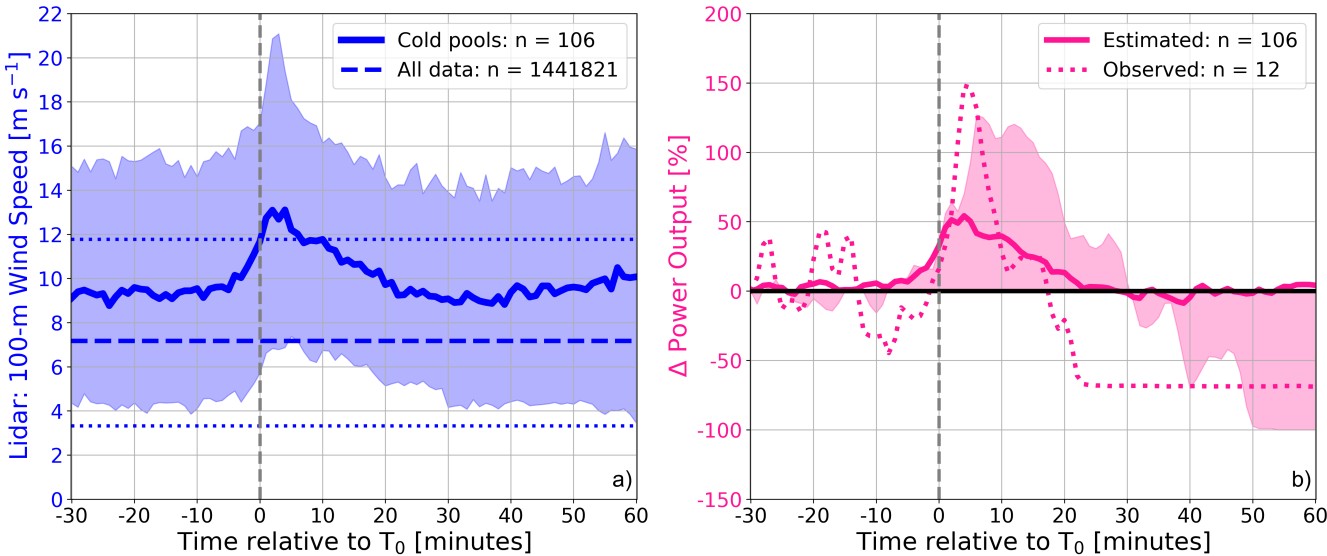

**Figure 9.** Time series relative to $T_0$ for (a) 100-m lidar wind speed (m s$^{-1}$; dark blue) and (b) power output change relative to $T_0$-30 minutes (%; pink). Solid lines indicate (a) median wind speeds around cold pool time periods and (b) median estimated power change. Thicker dashed lines indicate (a) 100-m climatological median wind speed and (b) observed median power change for the westernmost turbine, with shading in (a, b) and thinner dashed lines in (a) indicating the 10-90th percentile spreads. Sample sizes given in the upper right.

## 6 Conclusions

With the increasing amount of wind energy in Germany and remaining uncertainties regarding wind power prediction, it is crucial to better understand how particular atmospheric phenomena can impact wind turbines. Past literature has identified deep convection as one such phenomena linked to wind power prediction errors (Steiner *et al.*, 2017), with the associated convective outflow (i.e., cold pool) causing rapid changes in boundary-layer winds that are notoriously difficult for conventional forecasting models to characterize with sufficient spatial and temporal accuracy. To this end, we quantify the impact of convective cold pool passages on short-term environmental variations near two utility-scale wind turbines in northern Germany at DLR's WiValdi research wind park. We make extensive use of 1-minute meteorological mast and remote-sensing vertical profiles (Doppler wind lidar, MWR) in order to reliably capture the minute-scale ramp-up of thunderstorm winds (Lombardo *et al.*, 2014) and associated thermodynamic fluctuations. Cold pool events are identified from ground weather station data using thresholds for a decrease in $\theta_v$, positive daily wind anomaly, and measurable rainfall following from the methodologies of Kirsch *et al.*(2021) and Kruse *et al.*(2022). Of note, we define '$T_0$' as a proxy for the start of the cold pool passage similar to Kirsch *et al.*(2021), though the actual cold pool leading edge would precede this time by a few minutes due to the chosen threshold criterion. With the 120 detected cold pool cases across 4 years, we primarily focus on kinematic and thermodynamic changes within the turbine rotor layer (34-150 m) associated with the cold pool leading edge (i.e., gust front) and the subsequent thermal stabilization within the cold pool interior.





A typical WiValdi convective cold pool follows the quintessential sequence of environmental changes: surface wind speeds increase and the wind direction shifts a few minutes before a drop in surface $\theta_v$, which is then followed by rainfall 5-10 minutes later that peaks at 2 mm hr$^{-1}$. Of greater importance for wind energy, the hub-height median wind speeds relatively increase by up to 4 m s$^{-1}$ compared with the pre-event flow (taken as $T_0$-30 minutes) during the time period from $T_0$ to $T_0$+5 minutes, with

a relative increase in wind speeds for about 30 minutes. The hub-height wind direction shifts more westerly on average than the background flow by +15°, with full recovery to the background wind direction not reached even after 1 hour. Turbulence strength, which we quantify using the dissipation rate of turbulent kinetic energy, similarly shows a relative increase around the gust front time period and a subsequent gradual decrease. After the kinematic variations associated with the passage of the cold pool gust front, the near-surface atmosphere cools for 20 minutes until reaching a minimum relative $\theta_v$ decrease of -2 K, with

a relative decrease of -0.5 to -1 K still remaining after 1 hour. We determine that the gust front ramp-up starts from $T_0$-5 to $T_0$ minutes and peaks from $T_0$ to $T_0$+5 minutes, while the minimum $\theta_v$ (i.e., maximum cooling), which would be approximately the cold pool center (Kirsch *et al.*, 2024), is reached by $T_0$+15 to $T_0$+20 minutes. This yields a time from the cold pool leading edge to the cold pool center on the order of 20-25 minutes. The peak wind speed and maximum cooling time periods are used for subsequent analyses to isolate the largest environmental changes associated with the cold pool passages.

Using available vertical profiles from a Doppler wind lidar for 106 of the 120 cases, we estimate a median cold pool depth of 650-700 m over which kinematic changes induced by the cold pool gust front are detectable. This is in good agreement with a median cold pool depth of 750 m found by Kirsch *et al.*(2021) using pressure perturbation extrapolation of meteorological mast data, and highlights that a typical cold pool depth would exceed the height of current wind turbines. As the lidar can not provide consistent coverage below 100 m height, we leverage meteorological mast instrumentation to provide greater vertical

coverage within the turbine rotor layer, though with a more-limited sample size. We find a 'nose' in increased wind speeds near hub height during the gust front passages in line with past work (Lombardo *et al.*, 2014; Gunter and Schroeder, 2015; Canepa *et al.*, 2020), and which also manifests in the $\theta_v$ profile. The relative wind direction and $\theta_v$ changes conversely maximize near the surface, though a shift in direction of +5-15° during the gust front and cooling of 1.5-2 K from $T_0$+15 to $T_0$+20 minutes is still observed on average throughout the rotor layer.

The environmental variations as the gust front impacts WiValdi produce rapid increases in relative wind shear of 1.5 m s$^{-1}$ and negative relative wind veer of 8° over the entire turbine rotor layer, with an increase in static stability of a few K km$^{-1}$. However, these changes are not uniform across the turbine, but are rather asymmetric. The relative changes below turbine hub-height noticeably exceed those above hub-height: upper-rotor-layer shear is negligible, with the relative wind veer and stability magnitudes being twice as large below hub-height compared to above hub-height. Also, while there is an increase in

stability below hub-height, there is a temporary decrease in stability above hub-height likely indicative of the cold pool leading edge vertical circulation. With these vertical asymmetries, we would expect temporarily opposite fluctuations in static stability along with differential turbine structural loads across the rotor plane during cold pool gust front passages. Gradually increasing stabilization over the following hour across the whole rotor layer would have longer-lasting implications for turbine wake lengths, which can significantly affect power output for wind park configurations. Similarly, as wind direction more slowly



shifts back towards the pre-event flow than wind speed, there would be longer-lasting effects on turbine yawing and rotor-layer veer for structural loading.

As the pre-event winds typically lie above the cut-in wind speed and below the rated wind speed, we expected that cold pools would produce some wind power variations. To better quantify the potential power changes, we use the lidar hub-height wind speeds, a linear wind speed-power relationship within the partial load region of the power curve, and the operational 455 settings of the WiValdi turbines. We estimate a median relative power increase of up to 50% associated with the cold pool gust front, with a positive change in estimated power lasting 30 minutes. Although with just a handful of available power cases, the observed power variations qualitatively support a gust front peak and increased power occurring over approximately 0.5 hours. The median estimated percentage change in power due to the detected cold pool cases is noteworthy, however, the real-world significance of such a power change depends upon a number of factors beyond the scope of this study, including 460 how many turbines are affected, whether the cold pool occurs during a power transition period (e.g., early evening), and whether cold pools affect whole wind parks, including changes in wake effects through stability and wind direction changes. As such, the determination of the importance of cold pools for observed wind power changes is left for future work. Additional investigation relating observed cold pool characteristics with parent convective cell characteristics is needed to reduce wind power forecasting uncertainties related to deep convection.

Throughout our analyses, we highlight the remarkable capabilities of a vertically-scanning Doppler wind lidar to capture both the temporal evolution and vertical structure of short-term wind speed and direction changes associated with convective cold pools in terms of both the median and 10th-90th percentile spread. Conversely, we note an underestimation of the hub-height $\theta_v$ variations by the 1-minute MWR zenith scans, particularly during periods of rainfall. Future studies should investigate whether rainfall corrections and/or different 1-minute MWR scanning patterns could remedy this limitation during convective 470 cold pool passages. Finally, while we have a sufficient sample size of detected cold pools to support our conclusions regarding the kinematic and thermodynamic variations at WiValdi, a larger number of wind turbines over a greater area are necessary to generalize our findings for wind energy applications.

## 7 Code availability

Currently, the code is not publicly available.

## 8 Data availability

Currently, the data is not publicly available.

## 9 Author Contributions

All authors conceived of the paper idea. JT created the cold pool detection methodology. NW and JT performed the lidar and microwave radiometer data collection and processing. GK performed the radar data collection and analysis. JT performed



all other observational data analyses, with constructive insights from GK and NW. The manuscript writing was primarily done by JT, with edits done by GK and NW.

## 10 Competing interests

The authors declare that they have no conflicts of interest.

## 11 Acknowledgements

We must particularly thank those that helped with the construction and implementation of the WiValdi research wind park, from where we obtained the majority of our data. We also acknowledge Martin Hagen for help with the microwave radiometer data collection and processing, as well as helpful comments on this manuscript.

## 12 Financial Support

This research was partially funded within the DFWind2 project (FKZ 0325936A) funded by the German Federal Ministry
for Economic Affairs and Climate Action (BMWK) based on a resolution of the German Bundestag.

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
