# Peer review of "How do convective cold pools influence the atmospheric boundary layer near two wind turbines in northern Germany?"

_Wind Energy Science, 2025_

## Referee Comment (RC1)

**Peer review of: How do convective cold pools influence the boundary-layer atmosphere near two wind turbines in northern Germany?**

**Author(s): Jeffrey D. Thayer et al.**

**MS No.: wes-2025-38**

**MS type: Research article**

https://wes.copernicus.org/preprints/wes-2025-38/

**General Comment**

The manuscript "How do convective cold pools influence the boundary-layer atmosphere near two wind turbines in northern Germany?**"** addresses a timely and relevant topic which bridges atmospheric sciences with wind energy: convective cold pools (CPs). The downdrafts produced by convective rain have acquired renewed interest in the last 3-4 years, with several observational studies published characterizing composites of CPs from meteorological tower measurements, and from networks of ground-based weather stations. This study by Thayer et al. is the first to my knowledge that focuses on the characteristics of CPs specifically in regards to the effect on wind turbines (in the context of the WiValdi test site), focusing on the heights where the rotor is located. The data is new, the text is very clear, the methods sound, and the figures explicative. I find this work ready to be published, after taking into consideration minor comments, listed below.

**Specific Comments**

Format: Line – comment

85 – maybe say "Northern Germany" instead of Hamburg, since the two Kirsch studies (2021 and 2024) were conducted in Hamburg and Lindenberg (near Berlin) respectively.

94 – I'm not entirely sure but is there a typo here? Section 2 outlines are observational datasets → Section 2 outlines observational datasets

96 — I would mention that Section 5 also contains estimates of wind power increase.

142 – Would it be possible to make a table with the different instruments, the measurement resolutions, and what they were used for? Just to have a complete picture of the measurement set-up.

160 – What exactly is a positive daily wind anomaly?

170 – Should this say "A theta_v drop of *at least* 1.5 K occurs within 30 minutes of T_0" ?

175 – What does it mean that you prescribe that theta_v must recover at least *somewhat*? In case a person wanted to recreate your detection algorithm, what quantitative criterion would they have to include?

180 – What exactly is $t_0$ - 30 minutes? The 1-min minute averaged value of a variable at $t_0$–30 minutes, or the instantaneous value at $t_0$ - 30 minutes? Or the 30-min averaged value calculated between $t_0$-30 minutes and $t_0$?

186 – I would mention/quote here that Kruse et al (2022) linked convective cells to ground-based cold pools.

Table 1 – Maybe highlight "pre-event environment" in the table and specify in the caption. At a first glance, I thought these were the CP temperature drops and was very surprised.

Figure 2 – I do not see the magenta dot indicating WIVALDI in the plots.

Figure 3 – More a comment than a question: did you also look at specific humidity? This is the one feature of a CP that was not in agreement between the Kirsch 2021 and Kruse 2022 studies, with the Kirsch 2021 study showing an increase in moisture, and the Kruse 2022 study showing a decrease in moisture within the CP with respect to the pre-event environment. It would be interesting to see how moist/dry your CPs are since they are located towards a more coastal area like the Netherlands.

245-246 – I'm a little confused. Do you apply your detection algorithm to the temperature time series measured at 100 m and 85 m? Shouldn't you apply it to 2 m with the thresholds you used and check the corresponding temperature data at 100 m and 85 m? If you use the same threshholds at higher altitudes as for T2m, I would also expect that you find a decreased sample size at higher altitudes.

252 – Have you defined "sonic"?

272 – Then the Eddy dissipation rate "epsilon" rises again, right? Is this worth

310 – "averaged vertical profiles up to 1km height": What exactly does this mean? Could you add some words to clarify?

312 – Wouldn't you say that the median wind speed shows a relative increase from the pre CP environment up to about 700–800 m, from the plot?

319 – more-detailed → more detailed?

320 – more-limited → more limited?

320 – when → in which?

323 – "dashed black lines": In what plot?

373 – cut-outhub-height → cut-out hub-height?

Figure 8 – I might have missed this, but why is the sample size always different? Is it due to when the given sensors were active?

400-end (Conclusions) – The conclusions are written very clearly. I would however, like to see a paragraph that puts your WiValdi CPs into the context of the other CP composites measured in similar locations (Kirsch 2021 Hamburg, Kruse 2022 Netherlands, Kirsch 2024 Lindeburg); not only contextualising the detection method. One of course has to take into account that the detection methods are slightly different, since the detection algorithm you used is a bit tweaked with respect to the other Northen European studies, and that the locations are different (more coastal vs more in-land), which are details worth mentioning. Both the Kirsch 2021 and Kruse 2022 studies had measurements at "hub-height" (even if the focus was not on wind power) so there could be interesting comparisons there. This kind of contextualizing could also give an indication on whether the effects of CPs on wind power are expected to be the same everywhere, or completely different based on the location.

---

## Author Comment (AC1)

Peer review of: How do convective cold pools influence the boundary-layer atmosphere near two wind turbines in northern Germany?
Author(s): Jeffrey D. Thayer et al.
MS No.: wes-2025-38
MS type: Research article
https://wes.copernicus.org/preprints/wes-2025-38/

Comments

**Line 85** – maybe say "Northern Germany" instead of Hamburg, since the two Kirsch studies (2021 and 2024) were conducted in Hamburg and Lindenberg (near Berlin) respectively.

This is a good suggestion. We have changed this in the revised manuscript to "northern Germany".

**Line 94** – I'm not entirely sure but is there a typo here? Section 2 outlines are observational datasets → Section 2 outlines observational datasets

Thank you for catching this typo. We meant to say "Section 2 outlines **our** observational datasets." This has been changed accordingly in the revised manuscript.

**Line 96** — I would mention that Section 5 also contains estimates of wind power increase.

This is a good suggestion that will provide additional clarity for the reader. We have included it as follows in the revised manuscript: "Section 5 highlights changes throughout the turbine rotor layer and estimates of wind power increase during cold pool passages."

**Line 142** – Would it be possible to make a table with the different instruments, the measurement resolutions, and what they were used for? Just to have a complete picture of the measurement set-up.

We had originally decided against doing this due to the complexity of listing the different measurement systems in a single table, but have now endeavored to combine the data into a table as efficiently as possible. This instrument table now appears as Table 1 in the revised manuscript. We gladly would like any feedback on the understandability of this table.

We use 5 different types of data: Lidar, MWR, Inflow Mast, Ground Weather Station and Turbine. In the 1st paragraph of Section 2.1, we state both the starting date of measurements and what the instrument is used for. The measurement time periods for each instrument are given within Section 2.1 and in the new Instrument Table, and we explain the Lidar and MWR output frequencies being dependent on their respective scanning patterns.

In regards to the Inflow Mast data, we now include in the Instrument Table a list of each sensor *that we use in the manuscript* with their respective height, measuring time period, and output frequency. The use of these sensors to measure a certain variable is quite straightforward we feel, so we opted to not put the "use" in the table as well. Especially when calculating the virtual potential temperature, we require the combined use of the temperature, humidity, and pressure sensors, which would have complicated the table even more if we specifically listed their "use".

In reality, the inflow mast contains even more sensors beyond what are used in the manuscript that measure the atmospheric state variables (temperature, humidity, pressure, wind) at various heights from 2-149m. Specifically, we have 10 ultrasonic anemometers, 10 cup anemometers, 7 temperature sensors, 7 humidity sensors, 2 barometers, and 1 rain gauge. We do not have a sensor of each type at each height, but in general, we have these sensors at 10m, 34m, 62m, 85m, 120m, and 143m.

**Line 160** – What exactly is a positive daily wind anomaly?

This phrasing could be misleading, we agree. This phrase means a positive wind speed anomaly relative to the daily mean wind speed. In other words, this means that the wind speed around the time period of a cold pool gust front is larger than the average for that calendar day. We applied this criterion so that we detect gust fronts that stand out from the background wind conditions on a given day. We have re-phrased this in the revised manuscript as follows:

*"Continuous-$\theta_v$-decrease time periods include at least one time step of measurable rainfall exceeding 1 mm hr$^{-1}$ and a positive wind speed anomaly relative to the daily mean wind speed within +/- 10 minutes*

*of $T_0$. This rainfall threshold is used to remove instances of very weak convection or possible rainfall measurement error, while the wind speed anomaly is inspired by Kruse et al. (2022) and verifies a more significant cold pool gust front strength compared with the background flow conditions (given our interest in quantifying cold pool impacts on wind turbines)."*

**Line 170** – Should this say "A theta_v drop of at least 1.5 K occurs within 30 minutes of T_0" ?

Yes, you are correct. 'At least' has been added into the revised manuscript:
"A $\theta_v$ drop of **at least** 1.5 K occurs within 30 minutes of $T_0$."

**Line 175** – What does it mean that you prescribe that theta_v must recover at least somewhat? In case a person wanted to recreate your detection algorithm, what quantitative criterion would they have to include?

We could have explained this better, and we agree that we should describe this more quantitatively for reproducibility purposes. We have re-worded this sentence as follows:
*"Finally, we prescribe that $\theta_v$ must increase after reaching its minimum value and that this increase occurs within 60 minutes of $T_0$."*

**Line 180** – What exactly is t₀ - 30 minutes? The 1-min minute averaged value of a variable at t₀–30 minutes, or the instantaneous value at t₀ - 30 minutes? Or the 30-min averaged value calculated between t₀-30 minutes and t₀?

We agree that this could have been stated more clearly. We use the 1-min averaged values at $T_0$-30 minutes to represent the pre-event environmental conditions. The text has been amended to provide additional clarification:
*"We conservatively define the 'pre-cold pool environment' as the environmental conditions present at $T_0$-30 minutes, with findings by Kirsch et al. (2021) and Kruse et al. (2022) indicating that near-surface environmental conditions do not significantly change until at least $T_0$-15 minutes. The 1-minute averaged environmental conditions at $T_0$-30 minutes from the MWR, lidar, and mast data provide a proxy for the background environment prior to each of the detected cold pool passages."*

**Line 186** – I would mention/quote here that Kruse et al (2022) linked convective cells to ground-based cold pools.

Yes we should have done this before, since Kruse did use radar data as well. We have now cited Kruse et al. (2022) after point 1 in this sentence:
*"While weather station and meteorological mast observations can and have been used to identify convective cold pools (e.g., Kirsch et al., 2021; Hoeller et al., 2024), additional radar measurements can be useful to (1) confirm the presence of a parent convective cell linked to the ground-based cold pool (Kruse et al., 2022) and (2) provide comparison between radar-derived convection characteristics and near-surface cold pool characteristics."*

**Table 1** – Maybe highlight "pre-event environment" in the table and specify in the caption. At a first glance, I thought these were the CP temperature drops and was very surprised.

Thank you for this insight. We have now italicized the "pre-event environment" header within Table 1, and have included additional wording in the Table 1 caption to emphasize that these columns represent the conditions *before* the cold pool passages (and are not due to the cold pools themselves).

**Figure 2** – I do not see the magenta dot indicating WIVALDI in the plots.

Thank for you catching this mistake. We had forgotten to remove this phrase which corresponded to a previous version of Figure 2. We have removed this phrase in the updated Figure 2 caption, and highlighted that the figure panels are centered on WiValdi. We hope that with this information, in addition to noting the location of WiValdi in Figure 1, will be sufficient for the reader to understand where the location of WiValdi is in Figure 2.

**Figure 3** – More a comment than a question: did you also look at specific humidity? This is the one feature of a CP that was not in agreement between the Kirsch 2021 and Kruse 2022 studies, with the Kirsch 2021 study showing an increase in moisture, and the Kruse 2022 study showing a decrease in moisture within the CP with respect to the pre-event environment. It

would be interesting to see how moist/dry your CPs are since they are located towards a more coastal area like the Netherlands.

Yes, we agree that the specific humidity variations are an interesting point of discussion for cold pools. Kirsch et al. (2021) and Kruse et al. (2022) both seem to show decreases in relative specific humidity immediately after $T_0$. But then as you stated, they differ thereafter.

As shown below from our ground station attached to the MWR, we somewhat align with both Kirsch and Kruse. We have a longer-term decrease in relative moisture after $T_0$, in agreement with Kruse. However, we also show an increase in relative moisture from the pre-event environment after $T_0+25$ minutes or so, in agreement with Kirsch. Notably also, we show a little more than half of cases have a decrease immediately after $T_0$, but a sizable proportion also show an increase. Therefore, we would have to say that our observations do not wholly agree or disagree with either paper. As the near-surface moisture variations are not a crucial point related to the focus of our paper on the wind and stability changes within the turbine rotor layer, we decided to not highlight these observations in our paper. Nonetheless, it would be good for future research to delve deeper into this aspect of cold pools.

[Figure]

**Line 245-246** – I'm a little confused. Do you apply your detection algorithm to the temperature time series measured at 100 m and 85 m? Shouldn't you apply it to 2 m with the thresholds you used and check the corresponding temperature data at 100 m and 85 m? If you use the same thresholds at higher altitudes as for T2m, I would also expect that you find a decreased sample size at higher altitudes.

We do not use the detection algorithm at higher altitudes. The timestamps at which we detect cold pools are from the MWR ground station. Your 2nd question is exactly what we do. We use the timestamps from 2m, and then see at those times what different variables look like at other heights. For additional clarity, the 'decreased sample size of detected cold pool events' later in the same sentence is due to the Mast data availability since the Mast only started recording data in November 2022 (whereas the MWR started measuring in November 2020).

**Line 252** – Have you defined "sonic"?

This was overlooked. Thank you for catching this. "Sonic" refers to an ultrasonic anemometer for brevity, especially so that the legend labels in Figures 4 and 5 can fit the instrument names. This has been defined in Section 2.1 and changed afterwards to just "sonic" for consistency.

**Line 272** – Then the Eddy dissipation rate "epsilon" rises again, right? Is this worth

This is definitely an interesting feature. We believe that it is likely indicating an end to the near-surface stabilization (e.g., backside of the cold pool), where the near-surface environment is shifting towards the background atmosphere again. We think this is likely since we know that turbulence strength tends to be elevated under more convective conditions, and thus the near-surface environment is probably changing stability regimes by the time period where the eddy dissipation rate is increasing again.

However, as we sometimes had multiple cold pool events detected on a given day within a couple hours of each other that could influence conditions by $T_0+60$ mins (e.g., case study), and given our focus on the primary gust front passage which precedes the subsequent stabilization, and since we don't believe we have sufficient supporting evidence to identify the backside of the cold pool passage over WiValdi at this time, we chose to not describe this feature in the paper so as to not be too unduly speculative.

**Line 310** – "averaged vertical profiles up to 1km height": What exactly does this mean? Could you add some words to clarify?

We agree that the wording of this phrase could be better. This is referring to vertical profiles of wind speed and wind direction from our Lidar that extend up to 1 km height, and these 1-min vertical profiles are averaged from $T_0$ to $T_0+5$ minutes to better isolate the time period of the gust front passage. We have re-phrased this sentence as follows:

*"As the wind speeds induced by cold pools often maximize above the surface within the height range of wind turbines, we isolate the vertical structure associated with the peak gust front strength as shown in Figures 3 and 4 using vertical profiles up to 1 km height averaged from $T_0$ to $T_0+5$ minutes."*

**Line 312** – Wouldn't you say that the median wind speed shows a relative increase from the pre CP environment up to about 700–800 m, from the plot?

Yes, you are correct. We were trying to be too conservative with our description. The median relative wind speed increase has a zero-crossing point close to 800m, and the median relative wind direction change first crosses the zero point at roughly 700m. Therefore, we have edited the manuscript to reflect an estimated cold pool depth from these 2 metrics of 700-800m.

**Line 319** – more-detailed → more detailed?

This is perhaps a grammar preference, but we have changed it according to your suggestion in the revised manuscript.

**Line 320** – more-limited → more limited?

This is perhaps a grammar preference, but we have changed it according to your suggestion in the revised manuscript.

**Line 320** – when → in which?

This is a good catch. We are referring to the mast sample sizes in Figure 7. This has been added into the revised manuscript.

**Line 323** – "dashed black lines": In what plot?

This becomes more obvious later in the sentence, but we agree that the Figure reference could be placed earlier in the sentence to provide better clarity. This has been done in the revised manuscript:
*"A typical gust front peaks in strength around the turbine hub-height at +3 m s$^{-1}$ (Fig. 7a), exhibiting a nose shape within the turbine rotor layer (34-150 m; dashed black lines) as has been observed by past work (Lombardo et al., 2014; Gunter and Schroeder, 2015; Canepa et al., 2020), with the vast majority of cases showing increased wind speeds from the background flow."*

**Line 373** – cut-outhub-height → cut-out hub-height?

Thank you for catching this. This has been changed in the revised manuscript.

**Figure 8** – I might have missed this, but why is the sample size always different? Is it due to when the given sensors were active?

Yes exactly. We explain at the beginning of the Figure 7 discussion that "There is a more-limited sample size when all mast instrumentation heights are available...". On the inflow meteorological mast, we have 10 sonics alongside 6 temperature and humidity sensors, which do not all have the same data availability unfortunately. The same mast instrumentation is used in Figures 7 and 8, where the sample size when all sonics are available is "n=25" [for wind shear and wind veer; Figs. 8a-b] and the sample size when all temperature and humidity sensors are available is "n=35" [for static stability; Fig. 8c]. As

we explained this sample size difference earlier for the same instrumentation, we did not think that it required additional discussion again for Figure 8.

**Lines 400-end (Conclusions)** – The conclusions are written very clearly. I would however, like to see a paragraph that puts your WiValdi CPs into the context of the other CP composites measured in similar locations (Kirsch 2021 Hamburg, Kruse 2022 Netherlands, Kirsch 2024 Lindenburg); not only contextualising the detection method. One of course has to take into account that the detection methods are slightly different, since the detection algorithm you used is a bit tweaked with respect to the other Northen European studies, and that the locations are different (more coastal vs more in-land), which are details worth mentioning. Both the Kirsch 2021 and Kruse 2022 studies had measurements at "hub-height" (even if the focus was not on wind power) so there could be interesting comparisons there. This kind of contextualizing could also give an indication on whether the effects of CPs on wind power are expected to be the same everywhere, or completely different based on the location.

This is a good suggestion. We have now included a few sentences to place our study in the context of Kirsch and Kruse's respective studies.

'*In terms of the bulk cold pool characteristics (strength, temporal evolution, vertical structure) related to kinematic and thermodynamic variations, we are in broad agreement with other European cold pool studies (Kirsch et al., 2021; Kruse et al., 2022). However, the 'nose' in $\theta_v$ near hub height is one cold pool feature also found by Kirsch et al. (2021) that is not observed by Kruse et al. (2022). As such, additional observational work is needed to determine if this aspect of the cold pool vertical structure is commonly found, since its impact on wind turbine wakes in terms of static stability is crucial for assessing the net impact of convective cold pools on wind power production in whole wind farms.*'

---

## Author Comment (AC2)

General Comments:

The manuscript entitled "How do convective cold pools influence the boundary-layer atmosphere near two wind turbines in northern Germany?" by Jeffrey D. Thayer and co-authors studies the statistical footprint of convectively induced cold-air outflows on thermodynamic and dynamical properties of the atmospheric boundary layer at a research wind park in northern Germany. By applying a custom detection algorithm and analyzing meteorological in-situ and remote-sensing observations, the authors characterize the temperature, humidity, and wind signals of 120 cold-pool passages within the rotor layer of wind turbines. Their findings suggest that cold pools can temporarily increase wind energy output by up to 50 %, whereas the associated wind fluctuations vary asymmetrically across the rotor layer and the increased near-surface static stability could impact the turbulence in the wakes of a wind turbine.

Overall, the manuscript is well-written, clearly structured, and guides the reader well through the different parts of the study. In the introduction, the current state of knowledge on this relevant and timely topic as well as the new contribution of the study are clearly outlined. The used experimental setup and analysis methods are described in a mostly understandable and transparent way. Although the results part covers several aspects of this research topic, it is presented in a balanced way and does not become lengthy. Moreover, the results are always discussed in the context of the current literature. The only criticism I could make is that I feel that the study could have gone a step further and be clearer on possible implications of the results for wind energy applications, which would support the significance of this work even further. Nevertheless, my overall assessment of this study is very positive and after addressing the minor comments listed below, I am happy to recommend the manuscript for publication in WES.

Specific Comments:

- Title: I suggest to use the more common term "atmospheric boundary layer" instead of "boundary-layer atmosphere".
  We think this is a good suggestion and have changed the title accordingly.

- Line 13: Please introduce Theta_v.
  Thank you for catching this. This has been changed in the revised manuscript.

- L14: I would go with the term "static stability".
  This is a good suggestion and has been changed accordingly in the revised manuscript.

- L19: Please clarify that the -2 K cooling refers to the near surface conditions.
  This is a good suggestion and has been changed accordingly in the revised manuscript.

- L46: "Limited work" by itself does not define a scientific gap in literature. What exactly do the existing studies miss?
  This is a valid critique, though we do see this sort of language used often in other studies. Nevertheless, we have clarified this sentence in the revised manuscript to highlight that the scientific gap related to wind ramps lies in a lack of detailed examination of convectively-induced wind ramps: 'Beyond this 'convective systems' wind ramp category broadly outlined in Steiner et al. (2017), we are not aware of additional work concerning wind power fluctuations induced by convective wind ramps over Europe.'

- L49: "Rarely": See previous comment.
  Yes, we should have provided more clarity when using this word. To the best of our knowledge, Canepa et al. (2020) may be one of the closest studies to our work, in terms of analyzing thunderstorm outflows with meteorological mast and lidar data in a coastal location. They also highlight changes associated with wind-energy-relevant variables (e.g., wind speed, turbulence intensity), and verify the

presence of thunderstorms (using satellite and lightning data) as we do. However, they focus only on 10 high-end thunderstorm outflow events (i.e., downbursts), while there are less intense thunderstorm outflow types that still have important impacts for wind turbine power output and structural loads. They detect cases only through wind speed time series and do not place much emphasis on whether the thunderstorms produce surface rainfall (they include both wet and dry downbursts), while we focus on the surface temperature changes to identify thunderstorm cold pools and prescribe that measurable rainfall must occur at WiValdi. We also have a distinct difference in terms of focusing on wind energy and providing turbine power observations, while they focused generally on how the thunderstorm outflows could impact engineering structures.

We have amended this sentence in the revised manuscript as follows:

*'For example, Lombardo et al. (2014), Gunter and Schroeder (2015), and Canepa et al. (2020) each analyzed strong thunderstorm outflow events (i.e., downbursts) using wind energy-relevant variables with a limited sample size (10 or fewer), but did not investigate a larger sample of cases or the broader range of thunderstorm outflow intensities that can occur.'*

- L103: Name the measurement heights of the mast since these are visualized e.g. in Fig. 7.

This is a good suggestion. We have now included the sensor heights in the revised text as follows:

*"One 150-m meteorological mast located 2 D in front of the westernmost turbine provides inflow conditions throughout the turbine rotor layer. On this 'inflow mast', ultrasonic anemometers (hereafter, sonic) lie at heights of 34 m, 62 m, 85 m, 120 m, and 149 m, while temperature and relative humidity sensors are located at 10 m, 34 m, 62 m, 85 m, 120 m, and 143 m. Barometers are only located at 10 m and 85 m, so pressure values at other heights are obtained via the hypsometric equation starting at 10 m."*

- L104: How high are the three meteorological masts?

As we do not use data from the meteorological mast array in between the turbines, we decided to remove this part of the sentence for succinctness and to reduce any confusion around what mast data we are using.

- Fig. 1a: I suggest adding the locations of some major cities and a distance scale for reference.

This is a helpful suggestion. We have added the nearby major cities of Hamburg, Bremen, and Kiel. The inclusion of Hamburg is especially useful now for understanding the relative proximity of the Kirsch et al. (2021) observations to our wind park. We decided to not include a distance scale since the latitude and longitude labels are already provided, and "distance" is kilometers would somewhat change throughout the map given the change in latitude.

- Fig. 1b: I would also mention in the text that the position of the MWR and lidar has changed over time.

Thank you for catching this. We have now mentioned this in the revised manuscript as follows:

*"We note that the lidar and MWR were slightly moved within the wind park every couple of years (Fig. 1b, blue squares), but were moved on the same days, kept adjacent to each other, and remained east of the easternmost turbine by approximately the same distance."*

- L129: For clarity, I suggest to mention that the microwave radiometer is a passive instrument.

This is a good suggestion. We have incorporated this in the revised manuscript as follows:

*"Adjacent to the lidar, there is a passive HATPRO G5 microwave radiometer (MWR; Rose et al., 2005) manufactured by Radiometer Physics GmbH that was installed on 26 November 2020 and which provides vertical profiles of temperature and humidity. With an associated ground weather station for obtaining surface environmental conditions, an MWR passively retrieves vertical profiles through usage of multiple brightness temperature measurements within the oxygen and water vapor absorption bands."*

- L151: Please introduce Theta_v.

Thank you for catching this. This has been changed in the revised manuscript.

- L161: What does "positive daily wind anomaly" mean? Does this refer to the maximum wind speed of the day, a positive anomaly relative to the daily mean wind speed, or something else?

    This phrasing could be misleading, we agree. This phrase means a positive wind speed anomaly relative to the daily mean wind speed. In other words, this means that the wind speed around the time period of a cold pool gust front is larger than the average for that calendar day. We applied this criterion so that we detect gust fronts that stand out from the background wind conditions on a given day. We have re-phrased this in the revised manuscript as follows:

    "*Continuous-$\theta_v$-decrease time periods include at least one time step of measurable rainfall exceeding 1 mm hr$^{-1}$ and a positive wind speed anomaly relative to the daily mean wind speed within +/- 10 minutes of $T_0$. This rainfall threshold is used to remove instances of very weak convection or possible rainfall measurement error, while the wind speed anomaly is inspired by Kruse et al. (2022) and verifies a more significant cold pool gust front strength compared with the background flow conditions (given our interest in quantifying cold pool impacts on wind turbines).*"

- L172: Both Kruse et al. (2022) and Kirsch et al. (2021) use a 20-minute period to detect temperature drops related to cold pool passages.

    This is a good catch. We were somewhat confused by the phrasing in Kirsch et al. (2021). This has been changed in the revised manuscript as follows:

    "*The 30-minute time constraint is similar to that of past work (20 minutes for Kirsch et al., 2021 and Kruse et al., 2022).*"

- L175: Can you specify "at least somewhat"?

    We could have explained this better, and we agree that we should describe this more quantitatively for reproducibility purposes. We have re-worded this sentence as follows:

    "*Finally, we prescribe that $\theta_v$ must increase after reaching its minimum value and that this increase occurs within 60 minutes of $T_0$.*"

- Section 3: As this section is rather long, I suggest to introduce sub-sections for a clearer structure.

    We agree that Section 3 is quite long relative to the other sections. Therefore, we have now separated Section 3 into two subsections constituting 'near-surface' and 'hub-height' environmental changes.

- L209: Please specify the exact time period instead of only the years.

    This is a good suggestion and has been changed accordingly in the revised manuscript.

- Table 1: Why does the table only show the pre-event data? I think that listing the actual cold pool signals would be more instructive for the reader.

    We did not originally include the pre-event data since the median time series of the relevant near-surface variables were being shown in Figure 3. However, we agree that comparisons with other work is easier with the monthly climatological values in Table 1. Therefore, we have now added the relative changes of virtual potential temperature, maximum rainrate, and wind speed as additional columns in Table 1. We keep the original columns since they provide necessary context for the 3 new columns and they allow for comparisons with relationships described in Kirsch et al. (2021) and (2024).

- Table 1: Does the measurement accuracy of the instrument allow to show the data with a two-digit accuracy?

    This is a good point that we did not originally consider. The ground weather station sensors are quoted as having an accuracy of 0.2-0.3 K or m/s. Therefore, we feel that rounding to 1 decimal place is reasonable.

- L218: What is the median temperature decrease over the cases? Here, showing the corresponding data in Table 1 would help (see earlier comment).

    The median decrease in $\theta_v$ at 2 meters over all detected cases from the MWR ground weather station is 2.7 K. As the cold pool event signals are now also included in Table 1, this value should be

more obvious and understandable for the reader. This is now mentioned throughout the manuscript, including in reference to this comment as follows:

"*With $T_0$-30 minutes being a proxy for the pre-event environment, we find a median relative decrease in $\theta_v$ of 2.7 K (red) during the cold pool passages that occurs in the span of ~20 minutes, starting at $T_0$-5 minutes and reaching a minimum around $T_0$+15 to $T_0$+20 minutes.*"

- Figure 2: The differently colored dots and outlines are very hard to see in the plots.
  This is a valid critique. We have now removed the polygon centroid dots (as they are not consequential for our analyses), have edited the Figure 2 caption to remove the magenta dot description since that was a holdover from a previous version of the plots, and have increased the linewidth of the outline for polygons that overlap with WiValdi to enhance the contrast with those polygons that do not.

- L267-269: Are these measurements taken at the inflow mast? This is relevant for the interpretation of the results and could also be clarified elsewhere.
  This is a valid critique. We only used in-situ mast measurements from the inflow mast. We did not use data from the 3 meteorological masts in between the turbines, but we show all masts in Figure 1 to show an accurate representation of the measurement structures present within the WiValdi wind park. Nevertheless, we have indicated in additional places throughout Sections 3-5 that the in-situ mast observations within the turbine rotor layer being used are from the inflow mast.

- Fig. 4: I assume that the central lines show the respective median but this is not indicated in the figure caption.
  Thank for you catching this omission. We have amended the Figure 4 caption to include this description.

- L290-291: The median temperature evolution at 85 m (Fig. 4d) shows a slightly weaker signal than near the surface (Fig. 3). It might be worth mentioning that this is consistent with previous studies (e.g. Kirsch et al., 2021).
  This is a great suggestion! We have incorporated this content into the revised manuscript as follows:
  "*This decrease in median mast $\theta_v$ at 85 m (Fig. 4d) shows a slightly weaker signal than near the surface (Fig. 3), which is consistent with previous studies (e.g., Kirsch et al., 2021).*"

- L294-305: This paragraph feels a bit disconnected from the rest of the section. I am not entirely sure that the purpose of the case study is, apart from demonstrating that the composite properties of a cold pool also materialize in a single case. If the authors decide to keep the case study, I suggest to move it to an earlier location in the section, maybe in connection with Fig. 2.
  This is a good suggestion, especially given that we are showing a case study for the convection snapshot shown in Figure 2c. We have moved the case study to now be Figure 3. This then allows the reader to understand how different variables fluctuate with time during cold pool passages before we show composite statistics in Figures 4 and 5.

- L312: I would say that the wind speed perturbation reaches up to 800 m rather than 650 m.
  Yes, you are correct. We were trying to be too conservative with our description. The median relative wind speed increase has a zero-crossing point close to 800m, and the median relative wind direction change first crosses the zero point at roughly 700m. Therefore, we have edited the manuscript to reflect an estimated cold pool depth from these 2 metrics of 700-800m.

- L336: I am not convinced by this argument. I am inclined to think that the "nose" in the profile at hub height points to an impact of the rotor itself, but I can only guess what process causes this profile. This would be very interesting to know but probably involves some speculation.
  Yes, we agree that this statement is somewhat speculative. However, as the measurements in Figure 7c come from the inflow measurement mast, which is 2D upstream (to the west) of the western turbine (and given that rotor layer winds associated with these cold pool events predominantly come from a ~westerly direction), these measurements emanate from the boundary-layer flow that has not yet

impacted or interacted with the WiValdi turbines. If you are referring more to potential blockage effects upstream of the western turbine that could cause such a 'nose' feature in $\theta_v$, we believe that the inflow mast is sufficiently upstream enough that blockage effects are quite negligible and would not cause this 0.5-1 K difference between the hub-height and the rotor bottom/top heights. As such, we do not think that the turbine rotor would be the cause of this 'nose' in $\theta_v$.

We think it is more likely that given the nose in wind speeds shown in Figure 7a, this part of the cold pool gust front protrudes more into the warmer air out ahead of the propagating cold pool. And so, with the expected enhancement of mixing in the turbulent nose of the gust front, we believe that the hub-height air could be mixing more with the warmer air to produce less of a relative decrease in $\theta_v$ than above and below hub-height. The median $\theta_v$ time series for all cold pool cases (below) shows all Inflow Mast temperature sensor heights (10, 34, 62, 85, 120, 143 m) with the 2-m Ground Weather Station $\theta_v$, and confirms that the hub-height $\theta_v$ (green line) from $T_0$ to $T_0+5$ mins does not decrease as quickly as the temperatures at other heights within the rotor layer (i.e., 34-150 m). Before and after the gust front passage, the 85-m $\theta_v$ more closely lies in the middle of all heights. Therefore, we must conclude that something different occurs for the thermal vertical profile due to the gust front.

[Figure]

The schematic given below from Goff (1976) provides a helpful visualization of the gust front features we are trying to describe above. Do you perhaps have additional ideas about what could cause this 'nose' feature in temperature?

In any case, these sentences have been rewritten in the manuscript as follows to make it clearer that this finding is speculative:

*"The 'nose' in $\theta_v$ (Fig. 7c), with a lesser relative change at hub height compared to above and below, corresponds to the nose-like shape found in the wind speeds. Kirsch et al. (2021) find a similar 'nose' in $\theta_e$ at 110 m using mast observations at specific heights near Hamburg, however, this thermodynamic feature is not observed by Kruse et al. (2022) for cold pools over the Netherlands. We speculate that a $\theta_v$ 'nose' could perhaps be a manifestation of increased mixing in the gust front nose with the warmer background air ahead of the advancing cold pool, but we can not definitively identify the cause of this thermodynamic 'nose' feature, and so we leave this determination to future work."*

[Figure]

**Goff (1976)**

- L344-345: Could this be caused by different adjustment times of the temperature sensors between surface and mast (if the sensors are different)?

    The Met Mast temperature and humidity sensors (Hygro-Thermogeber-Compact 1.1005.54.441 sensor manufactured by Thies Clima) have a quoted response time of <20s, and the ground weather station attached to the MWR (Lufft WS600-UMB) is quoted as having a response time of <18s for 95% of the time. Therefore, we do not think the $\theta_v$ difference between the ground weather station and mast sensors is due to the adjustment time periods of the sensors.

    The median $\theta_v$ time series (above) shows the 2-m ground weather station decrease starting slightly after that of the mast heights. This is most likely due to the ground weather station being located approximately 1-km east of the Inflow Mast. Since the cold pool would predominantly impact the mast before the ground weather station, averaging from $T_0$ to $T_0+5$ mins yields a smaller relative decrease in $\theta_v$ at the ground weather station as shown in Figure 7c.

    We also must mention that there is warming which precedes the gust front passage (as the background flow ahead of the cold pool is lifted above the gust front nose), with this interaction complicating our "relative change" calculation. Nevertheless, it appears from the plot above that the near-surface temperature just decreases at a slower rate than the heights above, and that the near-surface temperature does eventually reach a lower minimum temperature than the mast heights within the rotor layer as would be expected. Therefore, we do not think that Figure 7c shows any feature out of the ordinary. It just comes down to the separation distance between the sensors.

- L400: Since section 6 mostly summarizes the methods and findings of the study, I suggest to call it "Summary & Conclusions".

    This is a good suggestion and has been changed accordingly in the revised manuscript.

- 470-472: I find the closing statement of the study rather weak. I would hope for more concrete implications for the wind energy applications to increase the significance of this timely and relevant study. Moreover, the authors could clarify what is exactly the benefit of an larger experimental setup compared to the current one.

    This is a very valid critique. We have added additional content in the last few paragraphs outlining more implications of this work, along with future suggestions for observational networks and analysis.

Technical Corrections:

- L102: Add space in "4.3 D".

    This has been changed accordingly in the revised manuscript.

- L190: What does "WN" mean?

  WN is the name of the DWD radar reflectivity dataset that we used. It is not an acronym, but rather is simply a name given to this particular radar reflectivity composite dataset by DWD that encompasses the time period of our study.

- L267: Add space in "subrange of".

  Thank you for catching this. This has been changed in the revised manuscript.

- Figs. 4, 5, 7, 8, 9: I suggest moving the labels a) to d) to the top left corner of the subplots for consistency.

  This is a valid critique. We have decided to place the figure subplot labels at the bottom right of each panel for consistency.

- L373: Add space in "cut-out hub-height".

  Thank you for catching this. This has been changed in the revised manuscript.